# FASTer: Toward Efficient Autoregressive Vision Language Action Modeling via neural Action Tokenization

**Yicheng Liu[1,4§]\*, Shiduo Zhang[2, 3§]\*, Zibin Dong[1,5]\*, Baijun Ye[1]\*, Tianyuan Yuan[1,4],**
**Xiaopeng Yu[2,3], Linqi Yin[2,3], Chenhao Lu[1,4], Junhao Shi[2,3], Luca Jiang-Tao Yu[6],**
**Liangtao Zheng[7], Tao Jiang[4], Jingjing Gong[3†], Xipeng Qiu[2,3†], Hang Zhao[1,4†]**

[1]Tsinghua University, [2]Fudan University, [3]Shanghai Innovation Institute, [4]Galaxea AI
[5]Tianjin University, [6]Hong Kong University, [7]UCSD

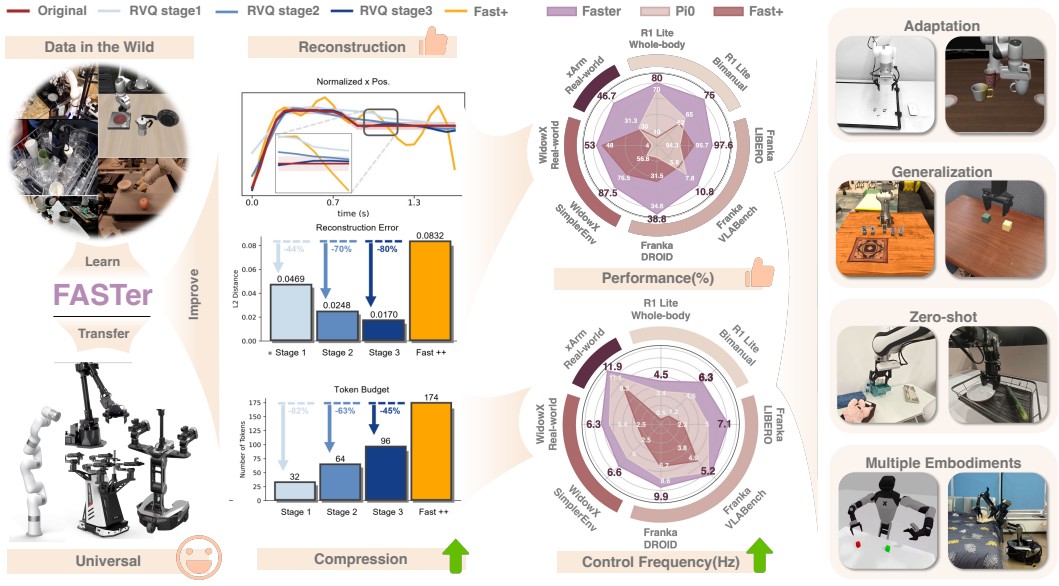

Figure 1: **FASTer** combines a learnable action tokenizer (FASTerVQ) and an autoregressive VLA model (FASTerVLA), achieving efficient compression, fast control, and strong performance across eight real and simulated embodiments.

## Abstract

Autoregressive vision-language-action (VLA) models have recently demonstrated strong capabilities in robotic manipulation. However, their core process of action tokenization often involves a trade-off between reconstruction fidelity and inference efficiency. We introduce **FASTer**, a unified framework for efficient and generalizable robot learning that integrates a learnable tokenizer with an autoregressive policy built upon it. FASTerVQ encodes action chunks as single-channel images, capturing global spatio-temporal dependencies while maintaining a high compression ratio. FASTerVLA builds on this tokenizer with block-wise autoregressive decoding and a lightweight action expert, achieving both faster inference and higher task performance. Extensive experiments across simulated and real-world benchmarks show that FASTerVQ delivers superior reconstruction quality, high token utilization, and strong cross-task and cross-embodiment generalization, while FASTerVLA further improves overall capability, surpassing previous state-of-the-art VLA models in both inference speed and task performance.

---

\*Equal contribution. †Corresponding authors. §Project leaders.
{liuyicheng23@mails, hangzhao@mail}.tsinghua.edu.cn
{sdzhang23@m, xpqiu@}.fudan.edu.cn, jjgong@sii.edu.cn

# 1 INTRODUCTION

Vision-Language-Action (VLA) models represent a paradigm shift in robotics, embodying generalist robot policies trained on increasingly large-scale robotic datasets (Chenjia Bai, 2024). These models are categorized primarily by their method of robot action prediction, with the most prominent approaches being diffusion-based (Team et al., 2024; Black et al., 2024) and autoregressive VLA (Belkhale & Sadigh, 2024; Kim et al., 2024; Pertsch et al., 2025; Zhou et al., 2025) models. While diffusion-based models have demonstrated superior precision in manipulation tasks, they often exhibit a notable deficiency in leveraging critical visual and linguistic cues (Pertsch et al., 2025; Dong et al., 2025). In contrast, recent research indicates that a carefully designed autoregressive VLA model can increasingly bridge the performance gap with its diffusion-based counterparts, while simultaneously offering enhanced instruction-following capabilities (Pertsch et al., 2025; Intelligence et al., 2025; Hancock et al., 2025), superior scene generalization (Pertsch et al., 2025), and effective transfer of common-sense knowledge (Brohan et al., 2023). Most importantly, autoregressive VLA models share the most architectural similarity to the highly successful Vision-Language Models (VLMs), suggesting significant potential for future advancements.

A pivotal challenge within autoregressive VLA models is the development of an appropriate tokenization scheme to discretize continuous robot action sequence into action tokens (Wang et al., 2025c; Pertsch et al., 2025). Numerous sequence modeling studies, including LLMs and Speech-LLMs, have demonstrated that tokenizer quality directly determines model performance (Radford et al., 2019; Zhang et al., 2023; Gong et al., 2025). A further challenge for autoregressive VLA models is their inference efficiency, as they are considerably slower than diffusion-based counterparts (Kim et al., 2024; Intelligence et al., 2025). From this perspective, an effective action tokenization method must fulfill four key requirements: i) **High compression efficiency**: It must generate a minimal number of tokens for long sequences to ensure efficient and fast inference. ii) **Robust reconstruction quality**: As fewer tokens reduce the available information space, it is essential to guarantee high reconstruction fidelity rather than pursuing compression at the expense of accuracy. iii) **2D structural modeling**: the tokenizer must account for the two-dimensional nature of action sequences—comprising both the *action dimension*, where each component carries distinct physical semantics with limited redundancy, and the *temporal (horizon) dimension*, which often exhibits substantial redundancy across time. Modeling these coupled structures jointly is essential to achieve a favorable trade-off between efficiency and accuracy. iv) **Flexibility**: The tokenizer should enable out-of-the-box applicability across different backbones, tasks, and embodiments, thereby serving as a measure of its generalization ability. However, our preliminary experimental results, as illustrated in Figure 1, indicate that existing tokenization methods fail to comprehensively satisfy these principles.

To address these challenges, we introduce **FASTer** (**F**lexible **A**ction **S**equence **T**okenization for **e**fficient inference), a unified framework consisting of two complementary components: FASTerVQ and FASTerVLA. **FASTerVQ** first non-uniformly groups the action sequence into semantically meaningful patches, effectively mitigating the imbalance of data distribution across different action dimensions. A hybrid transformer encoder is then employed to extract latent representations, which are subsequently quantized via residual vector quantization (RVQ) (Parker et al., 2025; Lee et al., 2022). This design achieves a superior balance between reconstruction fidelity and compression ratio—preserving fine-grained motion details even under a significantly smaller token budget. The tokenizer is trained to reconstruct actions in both temporal and frequency domains, enabling it to capture local dynamics and global trends simultaneously. Pretrained on large-scale robot datasets, FASTerVQ exhibits strong generalization across embodiments and tasks, achieving extremely high compression rates while maintaining near-lossless reconstruction, even for high-dimensional control such as whole-body coordination. Building upon FASTerVQ, **FASTerVLA** employs block-wise autoregressive decoding that leverages the structured latent space and the partial independence of tokens along the action dimension, enabling efficient parallel prediction within each block while maintaining coherent spatio-temporal structure. Furthermore, a lightweight action expert, architecturally aligned with the VLM backbone yet highly parameter-efficient, is introduced to bridge the modality gap between linguistic reasoning and continuous control. Together, these innovations allow FASTerVLA to achieve faster and more stable inference, stronger multimodal alignment, and consistent state-of-the-art performance across diverse tasks, embodiments, and model scales. Our contributions are summarized as follows:

- We propose **FASTerVQ**, a compact and high-compression-ratio action tokenizer that combines transformer-based residual vector quantization (RVQ) with a lightweight mixture mechanism. This design jointly compresses action sequences into a unified discrete codebook while preserving control-relevant structure, enabling downstream VLAs to more effectively leverage its outputs for action reasoning and generation.
- We introduce block-wise autoregressive decoding for efficient action-token modeling, together with a share structured action expert that aligns with the VLM backbone. These components jointly unleash the capability of autoregressive VLAs, allowing them to achieve faster inference while maintaining high accuracy.
- We establish a comprehensive benchmark covering four real robots and four simulated environments, providing the first systematic analysis of action tokenization for VLAs. Extensive experiments demonstrate that **FASTerVQ** achieves superior trade-offs between reconstruction fidelity and code length, and that **FASTerVLA** consistently delivers state-of-the-art performance across embodiments, tasks, and VLM backbones in both simulated and real-world settings.

## 2 RELATED WORK

**Vision-Language-Action models.** With advances in LLMs and multimodal systems (Sun et al., 2024; Achiam et al., 2023; Bai et al., 2023; Dubey et al., 2024), Vision-Language-Action (VLA) models have emerged as a promising direction for universal manipulation, leveraging the scalability of pretrained VLMs (Brohan et al., 2023; Driess et al., 2023; Kim et al., 2024; Black et al., 2024; Pertsch et al., 2025; Intelligence et al., 2025). Continuous control is typically modeled through either diffusion-based methods (Bjorck et al., 2025; Black et al., 2024; Kim et al., 2025) or autoregressive prediction over discretized actions (Brohan et al., 2023; Kim et al., 2024; Pertsch et al., 2025). Although autoregressive VLAs offer strong language grounding and generalization, they remain slower than non-autoregressive approaches (Brohan et al., 2023; Pertsch et al., 2025; Black et al., 2024). Our method instead learns a compact, structured discrete action space that aligns with pretrained VLMs, allowing autoregressive policies to operate at a more efficient token granularity and bringing inference speed into a practically usable regime.

**VQ Tokenizer.** VQ tokenization is widely used for modal compression across images (Yu et al., 2024; Sargent et al., 2025; Tian et al., 2024; Bao et al., 2022), video (Yu et al., 2023; Tang et al., 2024; Zhao et al., 2024; Wang et al., 2024a), audio (Casanova et al., 2024; Parker et al., 2025; Zhang et al., 2023; Lahrichi et al., 2025), graphs (Wang et al., 2025a; Zeng et al., 2025; Nguyen et al., 2024), and multimodal fusion (Sadok et al., 2025; Liu et al., 2025; Wang et al., 2024b). Audio codecs and action tokenizers share key traits: both process continuous time-series with short-term fluctuations, long-term trends, and periodic patterns (Parker et al., 2025; Zhai et al., 2025); both face non-uniform information density (Parker et al., 2025; Zhang et al., 2023); and both require strong temporal causality to ensure sequence coherence (Lahrichi et al., 2025). FASTerVQ therefore adopts an audio-inspired residual architecture, and our experiments confirm that, with sufficient data, such VQ designs inherit the strong generalization and scaling behavior observed across modalities.

**Action tokenization.** Autoregressive VLAs require discretizing continuous actions. Early approaches serialize each action dimension (Brohan et al., 2023; Kim et al., 2024), while later methods flatten full action chunks into 1D sequences (Zhao et al., 2023; Chi et al., 2024; Pertsch et al., 2025; Belkhale & Sadigh, 2024) using relative (Black et al., 2024; Intelligence et al., 2025) or delta actions (Pertsch et al., 2025). Existing tokenizers—binning (Brohan et al., 2023; Kim et al., 2024), DCT+BPE (Pertsch et al., 2025), and VQ variants (Belkhale & Sadigh, 2024; Wang et al., 2025c)—either under-compress, over-fragment, or lack reconstruction fidelity, producing action vocabularies that remain challenging for AR VLAs to model.

**Block-wise generation.** Chunk-wise generation appears across modalities: long-video models synthesize segments sequentially, conditioning each chunk on the last frame of the previous one (Zhang & Lim, 2024; Yang et al., 2025b), speech models adopt dynamic chunk-wise prediction for faster autoregressive synthesis (Li et al., 2025), and language models explore multi-token decoding or block-wise diffusion to reduce sequential depth (Li et al., 2024b; Arriola et al., 2025). In robot control, chunk policies (Zhao et al., 2023) are induced to mitigate compounding error, while (Zhang et al., 2025) show that chunked causal transformers further reduce autoregressive steps. (Kim et al., 2025) also performs parallel chunk decoding. Our approach instead performs block-wise autoregression over discrete action codes, enabling fast multi-token prediction while preserving AR consistency.

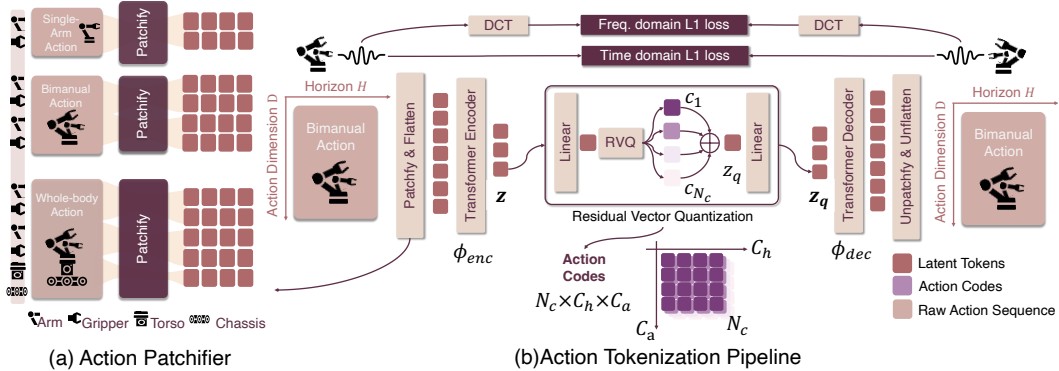

Figure 2: **FASTerVQ.** (a) The raw action sequence is patchified into compact tokens based on their embodied configuration. (b) FASTerVQ applies DCT and L1 reconstruction losses and adopts RVQ, encoding actions into $N_c$ code levels; each level can be reshaped into a $C_h \times C_a$ tensor.

## 3 METHOD

**Problem Formulation.** We consider the problem of learning a VLA policy that maps multimodal observations to sequences of robot actions. At each time step, the policy receives three inputs: an RGB image $I_t \in \mathbb{R}^{H \times W \times 3}$, a proprioceptive state $s_t \in \mathbb{R}^{d_s}$, and a language instruction $l \in \mathcal{L}$. The output is an action sequence of horizon length $H$, denoted as $A_{t:t+H} = (a_t, a_{t+1}, \ldots, a_{t+H-1})$. Instead of predicting continuous actions directly, we represent each action chunk as a sequence of discrete codes $(z_1, \ldots, z_N)$ obtained via a VQ tokenizer. The policy is then trained in an autoregressive manner to generate these codes conditioned on $(I_t, s_t, l)$. The VQ tokenizer subsequently decodes the generated codes back into continuous actions.

A key bottleneck of autoregressive VLA models lies in the token sequence length. Since tokens are generated sequentially, each requiring a full forward pass through a large transformer, inference latency grows quadratically due to attention complexity with respect to sequence length. Moreover, correct decoding requires all tokens to be predicted accurately, making the model brittle when token lengths vary. This issue is particularly acute in whole-body control, where high degrees of freedom lead to long action sequences. For example, the FAST tokenizer requires 150–200 tokens to represent a 2-second motion, which corresponds to an inference delay of roughly three seconds. Our experiments with policies initialized from multiple VLMs further confirm that training a VLA on variable-length codes is substantially more challenging than on fixed-length representations. To overcome these limitations, we propose **FASTerVQ** and **FASTerVLA**, which compress action representations, reduce the effective autoregressive depth, and equip the policy with a lightweight expert pathway attuned to action-level structure. The next sections unfold these components in detail.

### 3.1 FASTERVQ

Our action tokenizer architecture comprises two main components: an **action patchifier** and a transformer-based **Residual Vector Quantization (RVQ) Tokenizer**. This design leverages the unique characteristics of robotic action sequences to enhance tokenization efficiency and fidelity.

**Action Patchifier.** Although robot action sequences could be fed directly into a transformer as a sequential vector, a preliminary patching step proves to be more effective due to two key properties of these sequences. First, robot actions often exhibit smoothness and temporal redundancy, which can be addressed by chunk-wise grouping. This approach increases both the compression rate and the information density per token. Second, the different dimensions of an action vector correspond to distinct physical quantities with highly non-uniform data distributions. For instance, a robot's gripper state is often binary (open or closed), and the base movement is typically zero during arm manipulation. This distributional imbalance can pose training challenges. Grouping similar physical quantities a priori effectively mitigates this issue. Specifically, for an action sequence $A_{t:t+H} = (\mathbf{a}_t, \mathbf{a}_{t+1}, \ldots, \mathbf{a}_{t+H-1})$, where $\mathbf{a}_t \in \mathbb{R}^D$, we perform a two-dimensional partitioning. The temporal dimension is uniformly divided into $m$ groups of length $h$. And the action dimensions are non-uniformly partitioned into $n$ groups based on their physical characteristic (e.g., end-effector position, orientation, and gripper state are grouped separately). Each group is then padded to the largest group, $d$. This process yields a structured tensor of shape $(m \cdot h) \times (n \cdot d)$, which is subsequently flattened into a set of patches, $\mathbf{a}_{t:t+H}^P \in \mathbb{R}^{(m \cdot n) \times (h \cdot d)}$. This procedure can be conceptualized as a form of

non-overlapping convolution, a method widely adopted for early data processing in various domains due to its proven efficacy (Yu et al., 2023; Liu et al., 2023b).

**Residual VQ Action Tokenizer.** Inspired by the success of Residual Vector Quantization (RVQ) and transformer-based codecs (Parker et al., 2024), we design a transformer-based hybrid encoder–decoder architecture named Transformer Action AutoEncoder (TAAE). It combines the adaptive global receptive field of a transformer and the local relation modeling and downsampling capacity of convolutions, forming an effective information bottleneck. Moreover, RVQ naturally imparts a **coarse-to-fine** structure: early stages capture low-frequency components, while later stages refine high-frequency residuals. This property not only improves representational efficiency but also stabilizes both the training and inference of downstream VLA models. For an input action patch $\mathbf{a}_{t:t+H}^P$, the encoder $\phi_{\text{enc}}$ downsamples it into a latent embedding $\mathbf{z} \in \mathbb{R}^{C_h \times C_a}$. We then apply RVQ with $N_c$ quantization levels, decomposing $\mathbf{z}$ into residuals as $\mathbf{r}_1 = \mathbf{z}$, $\mathbf{r}_{i+1} = \mathbf{r}_i - \mathcal{Q}_i(\mathbf{r}_i)$, and the quantized latent embedding is $\mathbf{z}_q = \sum_{i=1}^{N_c} \mathcal{Q}_i(\mathbf{r}_i)$. Each quantizer $\mathcal{Q}_i$ selects its nearest codebook entry $e_k$ with

---

**Algorithm 1** FASTer Tokenizer

---

**Require:** FASTerVQ quantizers $\{\mathcal{Q}_i\}_{i=1}^{N_c}$, encoder $\phi_{enc}$ and decoder $\phi_{dec}$.
  **procedure** ENCODE($a_{t:t+H}$)
    $r \leftarrow z \leftarrow \phi_{enc}(\texttt{Patchify}(a_{t:t+H}))$
    $C \leftarrow [\,], z_q \leftarrow 0$
    **for** $i = 1$ to $N_c$ **do**
      $z_q \mathrel{+}= \mathcal{Q}_i(r), r \mathrel{-}= \mathcal{Q}_i(r)$
      $C$.append($c_i$)
    **return** $C$
  **procedure** DECODE($C$)
    $z_q \leftarrow 0$
    **for** $i = 1$ to $N_c$ **do**
      $z_q \mathrel{+}= \mathcal{Q}_i.\texttt{lookup}(C[i])$
    $a_{t:t+H} \leftarrow \texttt{UnPatchify}(\phi_{dec}(z_q))$
    **return** $a_{t:t+H}$
  **procedure** VQ_TRAINING($a_{t:t+H}$)
    $\hat{a}_{t:t+H} = $ DECODE (ENCODE ($a_{t:t+H}$))
    Train with the loss $\mathcal{L}$ in Equation (1)

---

index $k = \arg\min_j \|\mathbf{r}_i - e_j\|^2$. Collecting these indices yields a discrete code tensor $C \in \{1, \ldots, |\mathcal{Z}|\}^{N_c \times C_h \times C_a}$, where each element $c_{i,h,a}$ denotes the index of the chosen codebook vector at stage $i$ and location $(h, a)$. This tensor $C$ serves as the action tokens for the downstream policy. The quantized embedding $\mathbf{z}_q$ is then passed through the decoder $\phi_{\text{dec}}$ via a straight-through estimator (STE) (Van Den Oord et al., 2017) to reconstruct the action patch $\hat{\mathbf{a}}_{t:t+H}^P$.

**Training Objective.** The action tokenizer is trained by minimizing the reconstruction loss $\mathcal{L}_{\text{rec}}$ and the commitment loss $\mathcal{L}_{\text{commit}}$. The reconstruction loss $\mathcal{L}_{\text{rec}}$ is composed of two components: an $\ell_1$ loss on the temporal action signal $\mathbf{a}_{t:t+H}^P$ for step-wise action reconstruction, and an $\ell_1$ loss on the Discrete Cosine Transform (DCT) of the signal, $\text{DCT}(\mathbf{a}_{t:t+H}^P)$, to capture the overall trend. We select the $\ell_1$ loss due to its robustness against the extreme value noise often present in real-world robot action data, which contributes to training stability:

$$\mathcal{L} = \|\boldsymbol{a}_{t:t+H} - \hat{\boldsymbol{a}}_{t:t+H}\|_1 + \|\text{DCT}(\boldsymbol{a}_{t:t+H}) - \text{DCT}(\hat{\boldsymbol{a}}_{t:t+H})\|_1 + \lambda \cdot \|\boldsymbol{z} - \text{sg}(\boldsymbol{z}_q)\|_2^2, \quad (1)$$

where sg denotes the stop-gradient operation, and $\lambda$ balances the loss components. The RVQ codebooks are updated using an exponential moving average (EMA) with dead codes reinitializing.

## 3.2 FASTERVLA

**Architecture.** As shown in Figure 3 (a), FASTerVLA follows most VLM structure—a vision tower, a projection layer, and a transformer-based language backbone—to ensure compatibility with standard pretrained checkpoints. Visual inputs are encoded by the vision tower and projected into the model dimension; language instruction follows the usual text pipeline. *Action embeddings & Proprioception encoding:* We resize the original embedding table to include $|\mathcal{C}|$ new slots for action codes. Proprioceptive states are discretized into integers and tokenized as text. *Positional encoding and spacing augmentation:* We adopt rotary position embeddings (RoPE) for positional encoding. Since actions are encoded at a fixed sequence length, naively predicting a fixed-length target can lead to position overfitting. To mitigate this, we apply spacing augmentation: during training, the relative offset between adjacent action tokens is perturbed around unit spacing, i.e., if $p_i$ denotes the position of token $i$, we set $p_i = p_{i-1} + 1 + \epsilon_i$, where $\epsilon_i$ is a small integer jitter uniformly sampled from $\mathcal{U}[0, k]$ (e.g., $k = 2$), while preserving token order. At inference, we revert to a fixed spacing (e.g., $p_i = p_{i-1} + 2$). This encourages the model to rely on content rather than absolute positions within the horizon. *Lightweight action expert:* Inspired by $\pi_0$ (Black et al., 2024), we add a lightweight action expert sharing the backbone architecture but with fewer parameters. The backbone encodes the multimodal context once, while the expert autoregressively decodes action tokens from these

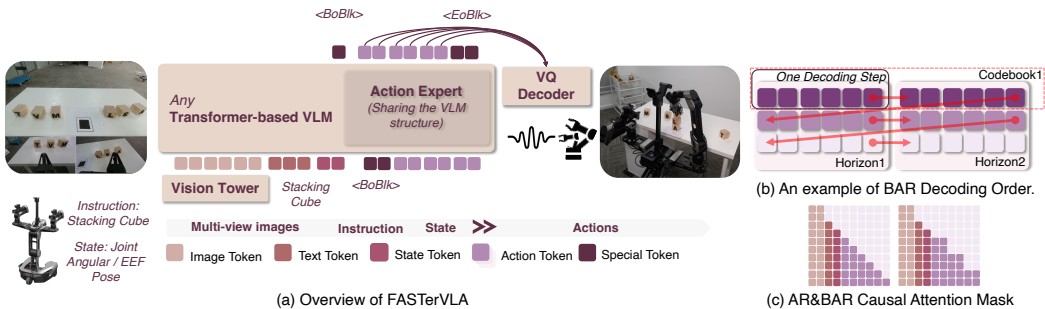

Figure 3: **FASTerVLA. (a)** The model takes RGB images, proprioceptive states, and a language instruction as inputs to a transformer–based VLM. An Action Expert autoregressively generates discrete action tokens, which a VQ decoder maps into the final continuous action sequence. **(b)** Codes are generated codebook–wise before advancing along the temporal horizon, yielding greater stability than horizon–first decoding (red arrows denote the decoding order). **(c)** Left: vanilla causal mask; Right: Block-wise causal mask, where tokens within each block attend to preceding and intra-block tokens. Token colors denote different modalities.

features. This design mitigates interference with pretrained weights while enabling lightweight and efficient decoding.

**Block-wise Autoregression (BAR).** In the vanilla setting, the policy is trained autoregressively with a next-token prediction objective under teacher forcing:

$$\mathcal{L}_{\text{AR}} = -\sum_{i=1}^{N} \log p_\theta\big(c_i \mid c_{<i}, I_t, s_t, x\big), \tag{2}$$

where $C = (c_1, \ldots, c_N)$ denotes the sequence of discrete action codes produced by the VQ tokenizer, and $N = N_c \times C_h \times C_a$ is the total number of tokens per action chunk. At inference, tokens are generated sequentially until the <eos> symbol, and decoded by the VQ decoder into a continuous trajectory $A_{t:t+H}$.

A key inefficiency of vanilla AR arises because many action codes are only weakly coupled across dimensions: distinct action dimensions often carry independent physical semantics and heterogeneous distributions. To address this, we adopt a block-wise objective that predicts the next block of tokens in a single forward pass. Specifically, we partition $C$ into $J$ contiguous blocks of size $B$, i.e., $C = \{C_1, \ldots, C_J\}$ with $C_j = (c_{j,1}, \ldots, c_{j,B})$ and $N = JB$. Training still uses teacher forcing, and the BAR loss is defined as

$$\mathcal{L}_{\text{BAR}} = -\sum_{j=1}^{J} \sum_{i=1}^{B} \log p_\theta\big(c_{j,i} \mid C_{<j}, I_t, s_t, x\big). \tag{3}$$

We replace the standard causal mask with a block-wise causal mask that allows intra-block attention (Fig. 3c). To integrate text and action generation, BAR uses two control tokens, $\langle\text{BoBlk}\rangle$ and $\langle\text{EoBlk}\rangle$, to toggle between block-wise and standard AR prediction, enabling seamless switching within a single sequence. In practice, when the model outputs $\langle\text{BoBlk}\rangle$, this token is replicated $B$ times and fed back as input to initiate prediction of the first block.

*Decoding order.* The VQ tokenizer outputs action codes arranged as a 3D tensor $C \in \mathbb{Z}^{N_c \times C_h \times C_a}$, spanning codebooks, temporal horizons, and action dimensions. FASTerVLA organizes decoding hierarchically across both the codebook and horizon dimensions (Fig. 3b). For each codebook, the model first decodes tokens along the horizon dimension $0, 1, \ldots, C_h - 1$ before proceeding to the next codebook. This ordering aligns with the residual quantization pipeline, where earlier codebooks capture coarse, low-frequency components, and later ones refine higher-frequency residuals. Consequently, decoding progresses in a coarse-to-fine manner, improving representational efficiency and stabilizing both training and inference. Since each block is decoded in parallel, BAR reduces the number of forward passes from $N$ to roughly $N/B$. As shown in Table 6, the number of blocks $J$ is small (e.g., 3 on LIBERO), yielding up to a $3\times$ reduction in inference latency compared to $\pi_0$ (Black et al., 2024).

## 4 EXPERIMENTS

We comprehensively evaluate the proposed **FASTer** framework through a series of experiments covering both the tokenizer and the full VLA model. We first investigate **FASTerVQ** to understand its

| Model | LIBERO | | | | | Simpler-Bridge | | | | |
|---|---|---|---|---|---|---|---|---|---|---|
| | Spatial | Object | Goal | Long | Average | Spoon | Carrot | Block | Eggplant | Average |
| Diffusion Policy (Chi et al., 2023) | 78.3 | 92.5 | 68.3 | 50.5 | 72.4 | - | - | - | - | - |
| Octo-Base (Team et al., 2024) | 78.9 | 85.7 | 84.6 | 51.1 | 75.1 | 12.5 | 8.3 | 0.0 | 43.1 | 16.0 |
| SpatialVLA (Qu et al., 2025) | 88.2 | 89.9 | 78.6 | 55.5 | 78.1 | 16.7 | 25.0 | 29.2 | **100.0** | 42.7 |
| $\pi_0$ (Black et al., 2024) | 96.8 | 98.8 | 95.8 | 85.2 | 94.2 | 66.7 | 58.3 | 58.3 | 88.3 | 66.7 |
| $\pi_{05}$ (Intelligence et al., 2025) | 98.8 | 98.2 | 98.0 | 92.4 | 96.8 | - | - | - | - | - |
| OpenVLA-OFT (Kim et al., 2025) | 97.6 | 98.4 | 97.9 | 94.5 | 97.1 | 12.5 | 4.2 | 8.3 | 0.0 | 6.25 |
| UniVLA (Bu et al., 2025) | 96.5 | 96.8 | 95.6 | 92.0 | 95.2 | 54.2 | 66.7 | 50.0 | 4.2 | 43.8 |
| OpenVLA (Kim et al., 2024) | 84.7 | 88.4 | 79.2 | 53.7 | 76.5 | 32.0 | 30.0 | 18.0 | 38.0 | 29.5 |
| Palligemma + Naive Tokenizer | 55.8 | 64.8 | 64.4 | 31.2 | 54.1 | 66.7 | 29.2 | 12.5 | 54.2 | 40.9 |
| MiniVLA (Belkhale & Sadigh, 2024) | - | - | - | 77.0 | - | 68.0 | 44.0 | **70.0** | 14.0 | 49.0 |
| VQ-VLA (Wang et al., 2025c) | - | - | 75.2 | 60.0 | - | 12.5 | 8.0 | 6.0 | 0.0 | 6.3 |
| $\pi_0$ FAST-R [1](Pertsch et al., 2025) | 96.4 | 96.8 | 88.6 | 60.2 | 85.5 | 29.1 | 21.9 | 10.8 | 66.6 | 32.1 |
| $\pi_0$ FAST-D (Pertsch et al., 2025) | 96.6 | 97.2 | 96.0 | 86.8 | 94.2 | 77.5 | 88.3 | 68.3 | 71.7 | 76.5 |
| FASTer w/o BAR | **99.4** | 98.8 | 94.8 | 88.6 | 95.4 | **97.5** | 83.3 | 65.0 | 78.3 | 81.0 |
| FASTer | 98.0 | **99.4** | **98.6** | **95.4** | **97.9** | 91.7 | **93.3** | 67.5 | **99.2** | **87.9** |

Table 1: Policy performance on Libero and Simpler-Bridge benchmarks.

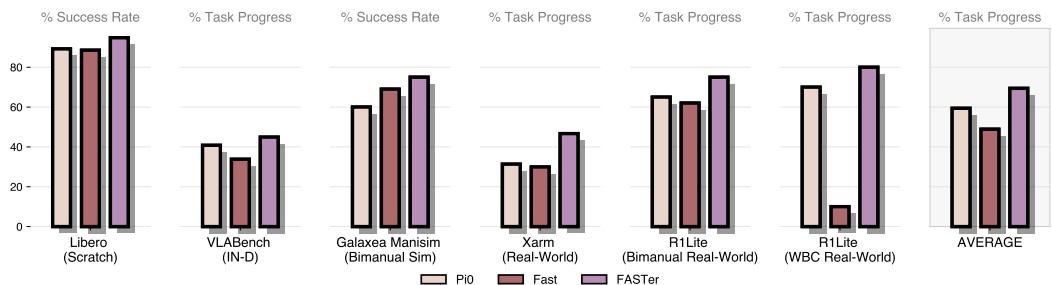

Figure 4: **Policy performance across embodiments and environments.** Results are reported in the in-distribution setting, covering two real-world embodiments and three simulated setups.

properties as an action tokenizer and then benchmark **FASTerVLA** across diverse tasks, embodiments, and backbones. Further ablation studies (4.4) analyze key design factors such as the tokenizer architecture and codebook size. These experiments aim to answer the following questions:
*1) How well does FASTerVQ balance accuracy and efficiency in modeling action chunks? (4.2)*
*2) How flexible is FASTer across tasks, embodiments, and backbones? (4.2 & 4.3)*
*3) How strong are the performance and generalization abilities of FASTerVLA? (4.3)*
*4) How do tokenizer performance and action code distributions affect overall VLA behavior? (4.3)*

## 4.1 EXPERIMENT SETUPS

**Benchmarks.** We evaluate across nine benchmarks spanning five distinct embodiments in both simulated and real-world settings as illustrated in Figure 11. These tasks include deformable manipulation, whole-body control, instruction following, and long horizon manipulation. Details of the benchmarks and the setup of each task are provided in Appendix A.1.

**Baselines.** We begin by evaluating FASTer on the most widely used public benchmark, comparing it against prior state-of-the-art models spanning both non-autoregressive (top) and autoregressive (mid) architectures in Table 1. As other experiments' baselines, we include $\pi_0$ (Black et al., 2024), a non-autoregressive VLA with flow matching, and $\pi_0$-FAST (Pertsch et al., 2025), the state-of-the-art autoregressive VLA built on the FAST tokenizer.

**Training.** Unless otherwise specified, all baselines and FASTerVLA models in our experiments are initialized from checkpoints pretrained on large-scale robotics data (e.g., from $\pi_0$-FAST). For Bridge and Droid experiments, all VLA models are instead initialized from pretrained VLM weights and pretrained on the same dataset to ensure a fair zero-shot evaluation. Detailed training configurations are provided in Appendix A.2.

---

[1] $\pi_0$ FAST-D: delta action chunks; $\pi_0$ FAST-R: relative action chunks. All results use $\pi_0$ FAST-D unless otherwise noted.

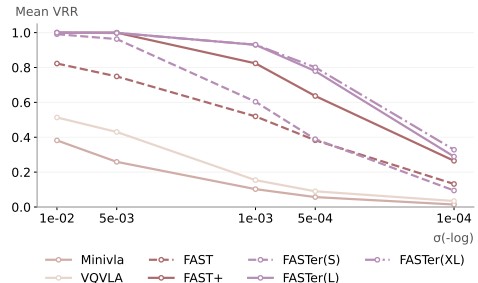
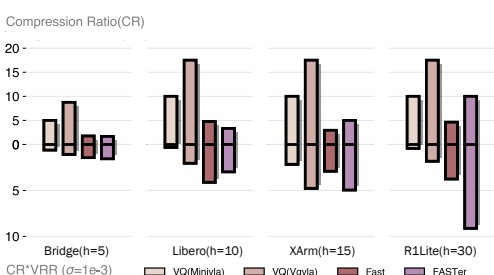

Figure 5: Average VRR of different action tokenizers under multiple error tolerances $\sigma$. FASTer achieves the best performance across all scales and exhibits clear data-scaling behavior.

Figure 6: Compression rate vs. reconstruction trade-off across action horizons, where FASTer achieves the best balance, particularly for long sequences.

### 4.2 RESULTS AND ANALYSIS OF FASTERVQ

As introduced in Section 1, an effective action tokenizer should jointly achieve high reconstruction quality and high compression rate. Figure 5 and Figure 6 compares FASTer against Fast and other VQ tokenizers in terms of compression efficiency and reconstruction fidelity. During training, we observed that reconstruction loss alone often fails to capture meaningful action quality, as many low-level discrepancies correspond merely to motor or sensor noise rather than task-relevant deviations. In such cases, perfectly reconstructing noisy signals yields little practical benefit, since actions with minor deviations typically result in indistinguishable outcomes during real-world execution. To better capture functional fidelity, we introduce the **Valid Reconstruction Rate (VRR)**, which measures the proportion of reconstructed actions whose deviation from ground truth falls within a small tolerance threshold:

$$\text{VRR} = N_{\text{valid}}/N_{\text{total}}, \quad N_{\text{valid}} = \sum_{i=1}^{N_{\text{total}}} \mathbf{1}\left(\left\|\mathbf{a}_i^{\text{recon}} - \mathbf{a}_i^{\text{gt}}\right\|_1 < \sigma\right), \tag{4}$$

where $N_{\text{total}}$ is the total number of actions, $N_{\text{valid}}$ counts those within tolerance $\sigma$, $\mathbf{a}_i^{\text{recon}}$ and $\mathbf{a}_i^{\text{gt}}$ denote reconstructed and ground-truth actions, $\mathbf{1}(\cdot)$ is the indicator function, and $\sigma$ is a hyperparameter. For robot end-effector translation, $\sigma$ corresponds to the Euclidean distance error measured in meters, whereas for end-effector rotation and joint positions, $\sigma$ represents an angular error in radians. In practice, a reconstruction error on the order of $10^{-2}$ is sufficient to cause a noticeable degradation in task execution accuracy.

**FASTerVQ Combines Compactness and Fidelity.** In Figure 5 and Figure 6, while baseline VQ methods perform well in terms of compression ratio, they exhibit significant shortcomings in reconstruction accuracy—even on in-distribution datasets, as shown in Figure 13. As a result, the LLM backbone learns inherently erroneous action codes, which constrain overall performance, especially in high-precision tasks. In contrast, FASTerVQ achieves a favorable balance between compression and fidelity, preserving reconstruction accuracy while minimizing the number of tokens. Figure 6 further compares tokenizer performance across different control frequencies and sequence lengths. Thanks to its structured design, FASTerVQ maintains significantly higher compression efficiency on high-frequency action sequences.

**FASTerVQ Efficiently Leverages Larger Data.** A competent action tokenizer is expected to be capable of effectively scaling with large amounts of training data. To evaluate this property, we trained three variants of FASTerVQ on different amounts of data—FASTER(S), FASTER(L), FASTER(XL), and the details can be found in Table 3. As shown in Figure 5, FASTerVQ consistently delivers state-of-the-art reconstruction across multiple error tolerances $\sigma$, e.g., FASTerVQ(S) vs. FAST and FASTerVQ(L) vs. FAST+. This capability clearly exhibits a strong data-scaling trend. In particular, FASTerVQ-XL achieves nearly lossless action-chunk reconstruction at the physically meaningful tolerance level of $\sigma = 10^{-3}$, even when FASTerVQ is trained with data budgets that are equal to or smaller than those of FAST. Furthermore, our results confirm that FASTerVQ's scaling behavior yields substantial gains in generalization—across tasks, embodiments and even action representations—demonstrated in Figure 8 and Figure 13. We will discuss this in detail in the following section.

**FASTerVQ Generalizes Across Embodiments.** We evaluate FASTerVQ's generalization by training the tokenizer solely on single-arm delta-EEF trajectories, and testing it on tasks that differ in

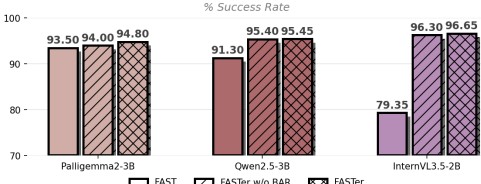 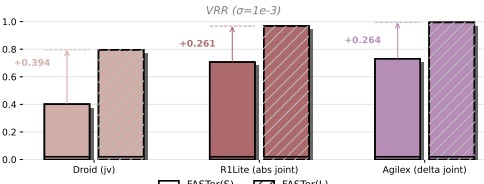

Figure 7: Comparison of FASTer, its non-BAR variant, and FAST **across backbones** on the Libero benchmark.

Figure 8: FASTer generalizes well **across embodiments** and **action types** with scaling trend.

semantics, embodiment, and even action representation. For unseen tasks and embodiments, we test on our newly collected Widow and XArm datasets (Figure 13). For unseen action representations, we evaluate joint-velocity actions from Droid, absolute joint-position actions from Galaxea Open, and delta joint-position actions from Agilex (Figure 8). Across all settings, FASTerVQ maintains strong reconstruction performance, suggesting that action chunks from diverse robot platforms share a common structure once mapped into normalized action space. This indicates that FASTerVQ captures a transferable action prior and shows a clear data-scaling trend in both cross-embodiment and cross–action-type generalization.

### 4.3 RESULTS AND ANALYSIS OF FASTerVLA

**In distribution Performance of FASTerVLA.** We first evaluate the performance of FASTerVLA and several SOTA baselines in an in-distribution (ID) setting. This setting is characterized by using the same tasks and scenes as in the training set, but with randomized object layouts to ensure robustness. The evaluation is conducted across a comprehensive set of task suites, including three simulation environments: LIBERO, VLABench, and GalaxeaManisim , and three real-world robotic platforms: single-arm using xArm, the bimanual using R1Lite, and whole-body control using R1Lite (Figure 11). Our results (Table 1, Figure 4) show that FASTerVLA achieves a 97.9% success rate on the Libero benchmark, setting a new state of the art. Across all evaluated settings, it consistently outperforms prior baselines, with only a marginal gap to $\pi_0$ on VLABench. These findings highlight the strong potential of autoregressive VLAs—when equipped with an efficient tokenizer like FASTerVQ—to rival and even surpass diffusion-based approaches.

**Out-of-Distribution Performance of FASTerVLA.** To evaluate the generalization and robustness of FASTerVLA, we conduct tests in an out-of-distribution (OOD) setting where the training and evaluation tasks are distinct. This evaluation spans both simulation and real-world environments, including the VLABench suite and two physical platforms: WidowX (with VLAs pre-trained on BridgeData (Walke et al., 2023)) and Franka (pre-trained on DROID (Khazatsky et al., 2024)). In simulation, FASTerVLA achieves the highest overall success rate while exhibiting the lowest relative performance drop (29%) compared to its in-distribution baseline (Figure 9). On the real-world robotic platforms, it consistently outperforms prior baselines (Figure 10). In particular, on the Simpler-Bridge benchmark, FASTerVLA attains an 87.9% success rate, outperforming the second-best model by 12.9%, further demonstrating its superior generalization capability.

To better understand these results, we analyze the code distributions of different tokenizers on BridgeData (Table 8). Fast+ (57% of 2048) and FASTerVQ (100% of 4096) exhibit markedly higher codebook utilization than Fast (48% of 2048). Moreover, Fast shows a dominant token with $F_{max} = 10\%$, while Fast+ and FASTerVQ achieve higher normalized entropy, indicating more balanced and information-rich token usage. Importantly, this balanced utilization translates into stronger task performance (Figure 10), where FASTerVLA and Fast+ achieve the highest task progress across zero-shot benchmarks such as *Bridge* and *Droid*. Overall, these findings suggest that diverse and evenly activated codebooks enhance action representation expressiveness, leading to better generalization and higher success rates.

**Cross-backbone ability of FASTer.** FASTer framework can also adapt to different backbones. We evaluated FASTerVLA with Palligemma (Steiner et al., 2024), Qwen2.5 (Yang et al., 2025a; Gao et al., 2025), and InternVL3.5 (Wang et al., 2025b) on the Libero benchmark (Figure 7). FASTer consistently improved performance, most notably raising InternVL3.5-2B's success rate by 17.3% to 96.65%, turning it from the weakest with FAST into the strongest overall. This gain stems from FASTer's concise, regularized, and data-driven representation, which better matches instruction distributions and avoids the inefficiencies of variable-length tokens in FAST. The figure shows

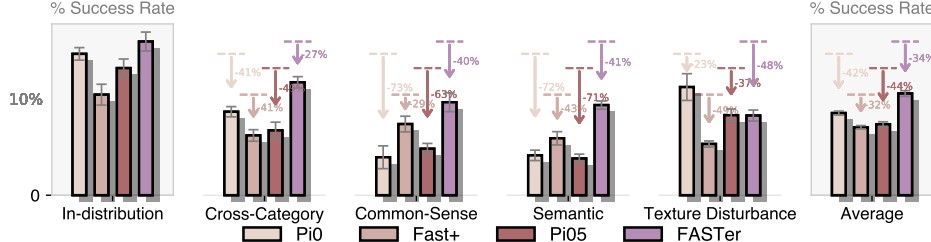

Figure 9: Evaluation results of generalization ability on VLABench. The evaluation dimensions cover generalization of language, vision, and goal and knowledge transfer.

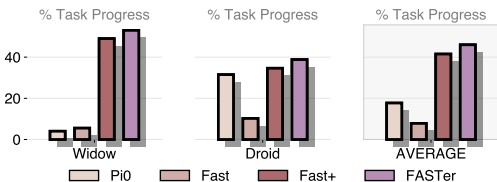

Figure 10: Comparison of zero-shot performance between FASTer and baselines on Droid and Bridge datasets.

| model part | Time(Single) | Time(WBC) |
|---|---|---|
| image encoders | 16 ms | 23ms |
| observation forward pass | 72 ms | 105ms |
| AR forward pass | 6.4*21 ms | - |
| BAR forward pass | 7.4ms*3 | 8.51ms*12 |
| FASTerVQ detokenization | 2.7 ms | 7ms |
| total inference | 112ms | 237ms |

Table 2: The inference time is measured using a PyTorch implementation on an RTX 5090. WBC refers to a whole-body-control setting that uses an action DoF/chunk size of 21/32, whereas Single refers to a single-arm setting that uses 7/20.

that FASTer's improvement is driven primarily by its neural VQ tokenizer: swapping FAST for FASTerVQ yields most of the gain, with BAR adding only a smaller incremental boost. Additional results are reported in Appendix A.4.

**Inference Efficiency of FASTer.** We evaluate the inference efficiency of FASTer on an RTX 5090, using a PyTorch implementation across the LIBERO and R1Lite–WBC benchmarks. In the Single setting, the single-arm configuration in the LIBERO environment uses an action chunk size of 20 with two camera views; in the WBC setting, the 21-DoF configuration uses an action chunk size of 32 with three camera views. A detailed analysis is provided in Table 2. The results suggest that the dominant bottleneck lies in the observation encoding stage: both $\pi_0$ and our model spend approximately 88–127ms on this step. By comparison, the action tokenizer remains relatively light, contributing only about 2.7–7ms. We also test an AR version of FASTer to compare with BAR. Because BAR emits multiple tokens per forward pass, it can meaningfully reduce latency when a task requires a large number of action tokens. Under the same environment and computational setup described above, we further compare **FASTerVLA**, $\pi_0$, and $\pi_0$–FAST with a shared PaliGemma backbone in Table 5. On LIBERO, the runtimes are 112ms for **FASTerVLA**, 176ms for $\pi_0$, and 197–556ms for $\pi_0$–FAST. In the R1Lite–WBC tasks, where the action dimensionality is substantially larger, **FASTerVLA** and $\pi_0$ converge to similar runtimes (around 230ms), since **FASTerVLA** performs 12 forward passes. In contrast, $\pi_0$–FAST incurs a significantly higher cost of 1,100–3,000ms, driven by the considerable number of action tokens required during decoding.

### 4.4 ADDITIONAL STUDIES

We ablate key design choices of **FASTerVQ** (tokenizer, codebook size, and residual depth) and **FASTerVLA** (action expert and block-wise decoding). The results confirm that the final configuration achieves the best balance of accuracy, efficiency, and stability. Detailed results and analysis are provided in Appendix A.3. Real-world rollouts of **FASTerVLA** are shown in Figure 15.

### 5 CONCLUSION

We propose an efficient autoregressive VLA framework that couples a flexible vector quantization module with a efficient VLA training inference setting. It delivers strong task performance with low-latency inference and generalizes across backbones, as the pretrained VQ can be reused in downstream tasks without retraining. The success stems from three key ideas: a codec-inspired RVQ balancing accuracy, diversity, and code length; block-wise autoregressive decoding for faster, expressive inference; and a lightweight mixture-of-experts VLA for action tokens. These results show that autoregressive modeling can be fast, transferable, and scalable for multimodal generation.

## 6 ETHICS STATEMENT

The authors have adhered to the ICLR Code of Ethics. This work does not involve human subjects, sensitive data, or raise any direct ethical concerns. All datasets used are publicly available.

## 7 REPRODICIBILITY STATEMENT

We are committed to ensuring the reproducibility of our research. Our source code will be made public upon publication. Detailed descriptions of our methods and model architectures are available in the section 3. All experimental settings, including datasets, training hyperparameters, evaluation settings are specified in the section 4 and Appendix.

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

# A APPENDIX

## A.1 DETAILED EVALUATION SETUPS.

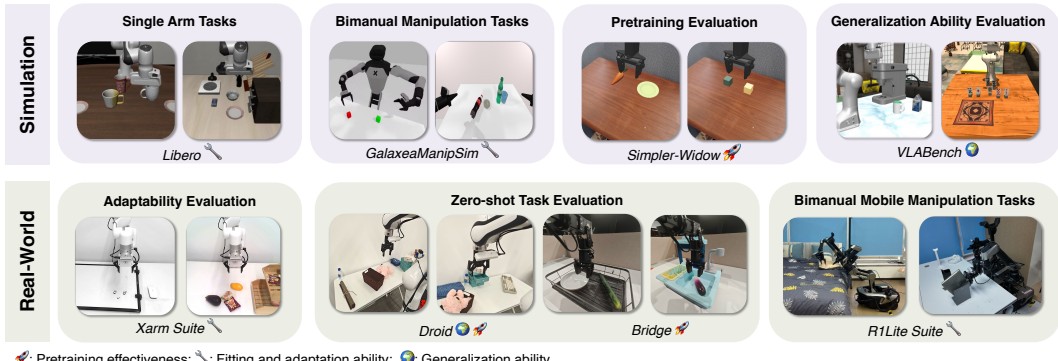

🚀: Pretraining effectiveness; ⚒: Fitting and adaptation ability; 🌐: Generalization ability

Figure 11: **Experiment setups.** Diverse embodiments, scenes, and tasks for assessing multiple capability dimensions (marked by emojis).

**Benchmarks&Evaluation Details.** We evaluate FASTerVLA across multiple dimensions, including out-of-the-box performance after pre-training, adaptability through fine-tuning on downstream tasks, and generalization ability under visual and language perturbations as well as in zero-shot settings. We adopt a variety of benchmark suites with distinct focuses, encompassing diverse robot embodiments and task distributions. For a fair ablation, we make every effort to use the same training settings as the other baselines. The evaluation setup below lists our training recipes and evaluation protocols.

- **Libero** (Liu et al., 2024) includes four task suites—Spatial, Object, Goal, and Long—with a total of 40 tasks, each containing 50 examples. This benchmark is particularly suitable for assessing both the models' capacity to fit the data and their adaptability to downstream tasks. To further compare the advantages of different model architectures, we evaluate two settings: models pre-trained and then fine-tuned on Libero, and models trained directly on Libero via imitation learning from scratch. We train a single policy across all four task suites. The results are reported in Table 1 and Figure 4, respectively. Following the official $\pi_0$ fine-tuning recipe, we train the model under a multi-task joint training setting for 30k steps, using a global batch size of 32, with image augmentation disabled, on 8 H100 GPUs. This model leverages FasterVQ(S), as listed in Table 3.

- **Simpler-widow** (Li et al., 2024a) serves as a benchmark for assessing the pre-training performance of VLAs. Table 1 reports the evaluation results of models trained from scratch on Bridge dataset (Walke et al., 2023). The models were trained for 4 epochs with image augmentation enabled, using a global batch size of 128 across 8 H100 GPUs. FASTerVLA utilizes FasterVQ(S), as described in Table 3.Unlike the default Simpler settings, which evaluate 24 trials for each task, we evaluate each task for 120 trials to mitigate the noise introduced by the high variance in zero-shot evaluation.

- **VLABench** (Zhang et al., 2024) is a benchmark focusing on the generalization capabilities of VLAs. We finetune our models on 10 tasks from VLABench, with a total of 5,000 examples. VLABench offers rigorously controlled evaluations across multiple dimensions, including complex instruction following, visual disturbances, language grounding, and commonsense transfer. We fine-tune the pretrained models—$\pi_0$, $\pi_0$-FAST, and FasterVLA—on 5,000 diverse episodes covering 10 tasks from VLABench. All models were trained for 30k steps with a global batch size of 32, no image augmentation, and executed on 8 H100 GPUs. The tokenizer used in FASTer was trained separately on the VLABench primitive dataset.

- **GalaxeaManipSim** (Team, 2025). Using this simulator, we evaluate the model on seven dual-arm tasks—BlockHammerBeat, BlocksStackEasy, ContainerPlace, DiverseBottlesPick, DualBottlesPickHard, EmptyCupPlace, and ShoePlace—with a total of 7,000 examples. The embodiments used in the simulator are R1 and R1Lite. We chose $\pi_0$, $\pi_0$-FAST as the baselines. All models were trained for 4 epochs with image augmentation enabled, using a global batch size of 128 across 8 H20 GPUs. We use three camera views, including the head camera and the left and right wrist

Table 3: Overview of data mixtures used for training FASTerVQ.

| Mixture Name | Datasets | Ratio | Application |
|---|---|---|---|
| Galaxea | Galaxea Open Dataset (Jiang et al., 2025) | 1.0 | Used for R1Lite Evaluation. |
| Droid Mixture | Droid (Joint Velocity) (Khazatsky et al., 2024) | 1.0 | Used for Droid evaluation. |
| FASTer(S) | Libero (Liu et al., 2024) | 5.0 | Used for libero, simpler and widow evaluation. |
| | Bridge (Walke et al., 2023) | 1.0 | |
| FASTer(L) | Libero (Liu et al., 2024) | 5.0 | Used for Xarm finetuning and tokenizer scaling analysis. |
| | Bridge (Walke et al., 2023) | 1.0 | |
| | Kuka (Kalashnikov et al., 2018) | 1.0 | |
| | Fractal (Brohan et al., 2023) | 1.0 | |
| | Droid(EEF) (Khazatsky et al., 2024) | 1.0 | |
| FASTer(XL) | Libero (Liu et al., 2024) | 5.0 | Used for tokenizer scaling and action representation analysis. |
| | Bridge (Walke et al., 2023) | 1.0 | |
| | Kuka (Kalashnikov et al., 2018) | 1.0 | |
| | Fractal (Brohan et al., 2023) | 1.0 | |
| | Droid(EEF) (Khazatsky et al., 2024) | 1.0 | |
| | Droid(Joint Velocity) (Khazatsky et al., 2024) | 1.0 | |
| | Galaxea Open Dataset (Jiang et al., 2025) | 1.0 | |

cameras. The FASTerVQ was trained on the Galaxea Open Dataset (Jiang et al., 2025) without post-training.

- **Xarm Suite** features an XArm dataset that covers five tasks, with 500 real-world episodes. These tasks involve flexible object manipulation, multi-goal selection, and long-horizon planning. The tasks include: board wiping, poker choosing, fruit placing, object managing, and towel folding. We conducted joint training on 500 samples across these tasks, using a total batch size of 128 with image augmentation, for 30k training steps. We employ FasterVQ(L) in this model, as shown in Table 3.

- **R1Lite Suite** includes three real-world tasks to evaluate our system: two desktop-level tasks— picking cubes and table bussing—and one whole-body control task, making the bed. The dataset sizes are 200 demonstrations for picking cubes, 200 demonstrations for desk organization, and 1,000 demonstrations for bed making. We chose $\pi_0$, $\pi_0$-FAST as the baselines. We train our model and baselines for 8 epochs, using a global batch size of 256 across 64 H20 GPUs. The FASTerVQ was trained on the Galaxea Open Dataset (Jiang et al., 2025) without post-training.

- **Bridge** (Walke et al., 2023) is a large-scale manipulation dataset with over 60k trajectories. Following its hardware setup, we use Widow250s for zero-shot evaluation after pre-training. The checkpoints used in this experiment are the same as those in Simpler.

- **Droid** (Khazatsky et al., 2024) contains over 90k real-world robotics trajectories. To ensure consistent evaluation, we follow its hardware setup and evaluate models in a zero-shot setting after pre-training. Following (Intelligence et al., 2025), we train on the Droid dataset for 3 epochs using 64 H200 GPUs, with a per-GPU batch size of 16, resulting in a total batch size of 1024, and adopt joint-velocity control.

## A.2 IMPLEMENTATION DETAILS

**Data Mixture.** In the experiments of Section 4, we trained FASTerVQ with different data mixtures. We report the detailed mixtures in Table 3.

**FASTerVQ.** We trained all policies using AdamW with a learning rate of $1 \times 10^{-4}$, weight decay 0.1, and $(\beta_1, \beta_2) = (0.9, 0.95)$. The learning rate followed a cosine decay schedule with 1,000 warmup steps. Gradients were clipped at norm 1.0, and training used mixed precision (bfloat16). The single-arm policy (8M parameters) was trained with a batch size of 512 sequences of length 21 tokens, whereas the full-body policy (13M parameters) was trained with a batch size of 2048 sequences of the same length. All models were trained on $8\times$ H100 GPUs, and training converged within 300k steps. We set the codebook size to $K = 4096$, the latent dimension to $d = 128$, and the commitment cost to $\beta = 0.25$. The detailed hyperparameters and settings used for each evaluation environment are provided in Table 4.

**FASTerVLA.** We trained the VLA using AdamW with a learning rate of $2.5 \times 10^{-5}$ and weight decay $1 \times 10^{-10}$. The learning rate followed a cosine decay schedule with 1,000 warmup steps, and training used mixed precision (bfloat16). The attention dropout was set to 0.0. For inference,

Table 4: Detailed Hyperparameter Configuration for FASTerVQ and BAR.

| Suite | Embodiment | Action DoF | BAR block size | Action chunk size | $C_a$ | $C_h$ | Number of codebook |
|---|---|---|---|---|---|---|---|
| Libero | Franka | 7 | 7 | 20 | 7 | 1 | 3 |
| Simpler | Widow250s | 6 | 7 | 10 | 7 | 1 | 3 |
| VLABench | Franka | 7 | 7 | 10 | 7 | 1 | 3 |
| GalaxeaManipSim | R1, R1Lite | 14 | 7 | 32 | 14 | 2 | 3 |
| Xarm Suite | XArm | 7 | 7 | 20 | 7 | 1 | 3 |
| R1Lite Suite | R1Lite | 21 | 7 | 32 | 21 | 2 | 3 |
| Bridge | Widow250 | 6 | 7 | 10 | 7 | 1 | 3 |
| Droid | Franka | 7 | 7 | 20 | 7 | 1 | 3 |

Table 5: Inference time across environments and models, measured on an NVIDIA GeForce RTX 5090 using PyTorch.

| Environment \ Model | FASTer | pi0 | FAST |
|---|---|---|---|
| LIBERO (ms) | 112 | 176 | 197–556 |
| R1Lite-WBC (ms) | 237 | 225 | 1,100–3,000 |

we employed top-$k$ sampling ($k = 50$) with temperature $0.8$. Block-wise autoregressive decoding was used, with a block size of $8$ for WBC and bimanual tasks, and $7$ for the single-arm setting. The hyperparameters used for action tokenization and BAR are listed in Table 4. We set the conditioning window to $1$ step, and the final actions were clipped to the range $[-1, 1]$.

## A.3 ABLATION STUDY

**FASTerVQ Ablations.** As shown in Table 6, FASTERVQ exhibits clear performance trends across the three ablated dimensions. The TAAE tokenizer delivers the strongest reconstruction fidelity, reflected by the lowest $\ell_1$ loss, and also reaches the highest success rate, surpassing both the CNN and Transformer variants. Increasing the codebook size improves representation capacity up to $4096$ entries, at which point success rate and utilization peak before degrading at $8192$, indicating the onset of codebook collapse. Finally, residual quantization yields the most stable gains with two to three residual layers, whereas deeper stacks introduce diminishing returns or mild instability. Together, these observations support the final choice adopted in our design: a TAAE tokenizer coupled with a $4096$-entry codebook and three residual codebooks, which jointly provide the most favorable balance between reconstruction quality and policy success rate.

**FASTerVLA Ablations.** We examine two essential components of **VLA**: the action expert (AE) responsible for action tokenization, and the block-wise autoregressive (BAR) decoding mechanism. Table 7 reveals that on **libero**, AE consistently enhances SR relative to the no-AE baseline, with pretraining providing the most pronounced gain (97.9 vs. 95.5). On **simpler-widow**, AE trained from scratch collapses (23.6), suggesting that gradients from the VLM offer only a faint signal for learning a high-quality action tokenizer. Pretraining therefore becomes indispensable, lifting SR to 87.9. These observations naturally motivate a two-stage training scheme: first pretrain AE within the VLM, then fine-tune jointly with partial freezing. This schedule accelerates convergence and markedly stabilizes optimization. For decoding, BAR substantially improves both accuracy and efficiency: SR rises from 95.5 to 96.7, while latency is reduced by more than $2\times$ (323,ms to 140,ms). When BAR is combined with AE, SR further increases to 97.7 at the same low latency, offering the most harmonious balance between precision and speed.

We also ablate different BAR block sizes. Our intuition is that BAR becomes effective when the action-token sequence exhibits a relatively stable length and when meaningful correlations are preserved across tokens. Under such conditions, BAR can be applied directly and remains reliable. We evaluate multiple BAR block sizes using the same FASTerVQ configuration with an action code length of 42. As shown in Appendix A.3, performance remains stable as long as the block size lies within a reasonable multiple of the action-dimension code length. When the block size matches this code length, the result is the strongest. Empirically, these findings echo our design intuition: BAR integrates seamlessly with FASTerVQ because FASTerVQ preserves the latent structure of action chunks, allowing BAR to exploit this structure for coherent block-wise generation.

Table 6: **Ablation on FASTERVQ** policy success rate in libero simulator (SR, %) and $\ell_1$ loss (lower is better);.

| (a) Tokenizer | | | (b) Codebook Size | | | (c) Residual | |
|---|---|---|---|---|---|---|---|
| Arch. | SR | $\ell_1$ | Size | SR | utilization | #Res. | SR |
| CNN | 96.2 | 0.0027 | 512 | 95.4 | 100 | 1 | 93.4 |
| Transf. | 95.3 | 0.0036 | 1024 | 93.2 | 100 | 2 | 95.5 |
| TAAE | 97.9 | 0.0021 | 4096 | 97.9 | 99.6 | 3 | 97.9 |
| – | – | – | 8192 | 96.3 | 95.1 | 8 | 96.6 |

Table 7: Ablation studies of FASTer. Left: ablation study of the action expert (**AE**) on **libero**; Middle: AE on **simpler-widow**; Right: BAR decoding. "Pretrain" indicates whether the model is initialized from a checkpoint pretrained on robotics datasets; others without this label are all initialized from such pretrained checkpoints.

| libero (AE) | | simpler-widow (AE) | | bar decoding | | |
|---|---|---|---|---|---|---|
| Variant | SR ↑ | Variant | SR ↑ | Decoding strategy | SR ↑ | Latency (ms)↓ |
| No AE | 95.5 | No AE | 75.6 | Token-wise (AR) | 95.5 | 323 |
| AE (no pretrain) | 94.8 | AE (no pretrain) | 23.6 | Block-wise | 96.7 | 140 |
| AE (with pretrain) | 97.9 | AE (with pretrain) | 87.9 | Block-wise + AE (FASTer) | 97.7 | 140 |

## A.4 DETAILED RESULTS OF DIFFERENT VLM BACKBONES

To evaluate the effectiveness and adaptability of our framework, we conducted comprehensive experiments on the Libero benchmark. We selected several popular VLM backbones: Paligemma-3B (Steiner et al., 2024), InternVL3.5-2B (Wang et al., 2025b), and various sizes of Qwen2.5 (0.5B, 1.5B, 3B) (Yang et al., 2025a). For Paligemma and InternVL, we utilized their original VLM checkpoints. For the Qwen2.5 models, we adopted the VLM checkpoints provided by (Gao et al., 2025), which fine-tuned the base LLM with a vision tower and projector on the LLaVa v1.5 dataset (Liu et al., 2023a). All these backbones were subsequently trained from scratch on the Libero benchmark. We compared their performance against the baseline FAST tokenizer.

The detailed results in Table 9 demonstrate that our framework achieves consistent and significant performance improvements across all tested model architectures and parameter scales. Notably, while Paligemma-3B delivered the strongest performance among all models using the FAST tokenizer (93.50 Average), our framework was able to unlock the latent potential of other VLMs. For instance, InternVL3.5-2B, which had the weakest performance with the FAST tokenizer (79.35 Average), was elevated to become the top-performing model overall (96.65 Average) when paired with our tokenizer. This is particularly significant given that InternVL 3.5 is inherently competitive on embodied benchmarks, indicating that our method successfully unleashes its latent potential for action generation. This remarkable turnaround not only proves the effectiveness of our framework but also highlights its broad adaptability and its ability to fully harness the capabilities of diverse VLM backbones.

Table 9: Detailed performance of different VLM backbones on Libero

| Model | Tokenizer | Spatial | Object | Goal | Long | Average |
|---|---|---|---|---|---|---|
| Qwen2.5-0.5B | FAST | **92.60** | 95.80 | 70.00 | 78.20 | 84.15 |
| | Ours | 90.40 | **99.00** | **71.60** | **87.00** | **87.00** (↑ 2.85) |
| Qwen2.5-1.5B | FAST | **96.00** | 92.80 | 87.20 | **85.80** | 90.45 |
| | Ours | 94.40 | **97.80** | **92.00** | 84.80 | **92.25** (↑ 1.80) |
| Qwen2.5-3B | FAST | **97.80** | 93.80 | 90.20 | 83.40 | 91.30 |
| | Ours | 96.40 | **99.40** | **92.00** | **94.00** | **95.45** (↑ 4.15) |
| InternVL3.5-2B | FAST | 92.60 | 95.20 | 77.20 | 52.40 | 79.35 |
| | Ours | **97.80** | **98.60** | **97.20** | **93.00** | **96.65** (↑ 17.30) |
| Paligemma2-3B | FAST | 93.40 | **98.80** | 95.00 | 86.80 | 93.50 |
| | Ours | **96.60** | 98.00 | **95.20** | **89.40** | **94.80** (↑ 1.30) |

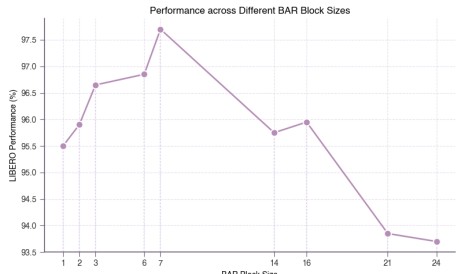

Figure 12: Different BAR block sizes, including irregular granularity, and their performance on the LIBERO environment.

Table 8: Comparison of vocabulary distributions of different tokenizers on the Bridge.

| Metric | Fast | Fast+ | FASTer |
|---|---|---|---|
| $N_{\text{vocab}} \uparrow$ | 2048 | 2048 | 4096 |
| $Usage$ [%] $\uparrow$ | 48.39 | 57.37 | 100.0 |
| $N_{\text{active}}(\sigma = 10^{-2}) \downarrow$ | 8 | 16 | 3 |
| $F_{max}$ [%] $\downarrow$ | 9.63 | 1.82 | 1.35 |
| $Entropy_{norm} \uparrow$ | 0.69 | 0.77 | 0.91 |

A.5 QUALITATIVE ANALYSIS OF THE TOKENIZER

To evaluate the effectiveness of different action tokenizers, we conduct a qualitative and quantitative analysis covering reconstruction fidelity, token utilization, and generalization across datasets and embodiments.

**VQ Settings.** The following VQ tokenizer configurations are used in our reconstruction visualizations and token usage analyses, covering prior baselines and our proposed model.

- **MiniVLA's VQ** (Belkhale & Sadigh, 2024) uses residual vector quantization to encode action chunks into sequences of codeword indices, with optional extra tokens in the vocabulary. For our experiments, we use the official VQ weights released by the authors: one trained on Libero and the other on Bridge V2 Dataset.

- **VQ-VLA's VQ** (Wang et al., 2025c) replaces OpenVLA's simple binning with a residual VQ tokenizer that encodes action sequences into discrete codeword indices using hierarchical quantization. In our experiments, we pretrain the VQ from scratch using the VQ-VLA training framework and code, on a 1:1 mixture of LIBERO and BRIDGE with a single H200 GPU (batch size 1024, 800k steps, learning rate 5e-5).

- **Fast+** (Pertsch et al., 2025)is the universal action tokenizer of Fast, trained on one million real-world robot trajectories.

- **FasterVQ** is our proposed VQ, with the training data mixtures summarized in Table 3.

**Action Reconstruction Visualization.** We qualitatively assess the learned VQ models by visualizing normalized action sequence reconstructions from Libero, Bridge, XArm, and Widow (Figure 14), comparing MiniVLA, VQ-VLA, Fast+, and Faster in terms of fidelity and generalization. In each plot, the red line denotes the ground-truth trajectory, the green line denotes the reconstruction, and the shaded red region denotes the error band with a tolerance of 0.01. Note that the green horizontal line in the figure appears due to clipping in the normalization at the 99th percentile bound.

On in-domain datasets such as Libero and Bridge, most predicted chunks remain within the 0.01 error band, indicating reasonable fidelity. However, their performance degrades notably on out-of-domain datasets like XArm and Widow, where a larger fraction of predictions exceed the tolerance. Reconstructions from MiniVLA and VQ-VLA exhibit more frequent spiky artifacts. Fast+ and FasterVQ preserve stable reconstructions across both seen and unseen domains, with most chunks remaining within the error band and some trajectories exhibiting nearly exact overlap with the ground truth.

**Token Activation Statistics.** Table 10 compares codebook size, active tokens, and usage frequency across VQ tokenizers. Small codebooks (VQ-VLA, MiniVLA) are fully activated. Fast+ underutilizes its larger vocabulary. In contrast, FasterVQ effectively harnesses a much larger codebook, activating thousands of tokens with a meaningful subset above frequency thresholds. This avoids token collapse and provides richer, more fine-grained representations.

**Detailed VRR Evaluation Results.** Figure 5 in Section 4 presents the average VRR values across four datasets for the FASTer, Fast, MiniVLA-VQ, and VQ-VLA-VQ tokenizers under different $\theta$ scales. Figure 13 provides a more detailed view, showing the VRR scores for each dataset at the three $\theta$ thresholds corresponding to meaningful physical error levels. Across multiple $\theta$ scales, the

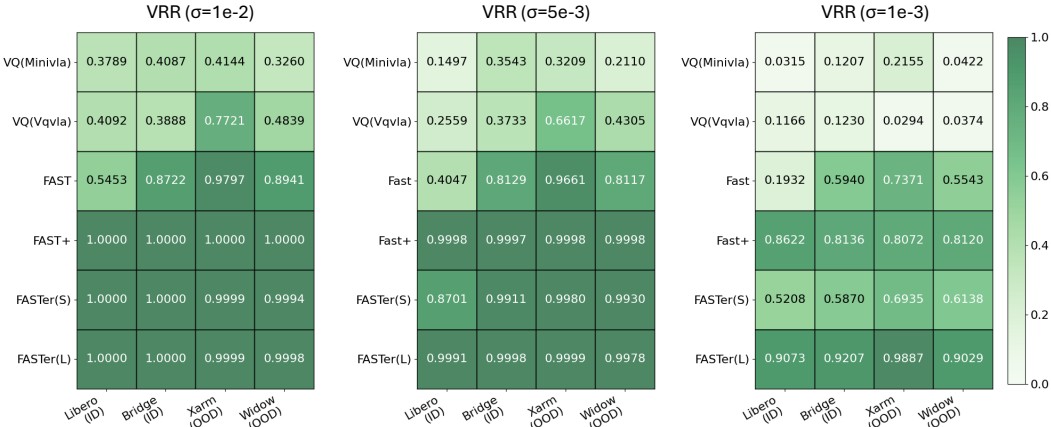

Figure 13: Heatmap for detailed VRR of different tokenizers across ID and OOD datasets.

Table 10: Token activation statistics on Bridge and XArm-mixture. Columns show codebook size, active tokens, and counts of tokens exceeding the $10^{-3}$ and $2 \times 10^{-2}$ thresholds.

| Tokenizer | Bridge | | | | XArm | | | |
|---|---|---|---|---|---|---|---|---|
| | CB | Act. | $> 1e^{-3}$ | $> 2e^{-2}$ | CB | Act. | $> 1e^{-3}$ | $> 2e^{-2}$ |
| VQ-VLA-vq (800k) | 256 | 256 | – | 1 | 256 | 256 | – | 1 |
| MiniVLA-vq-bridge | 256 | 256 | – | 1 | 256 | 256 | – | 1 |
| Fast+ | 2048 | 1175 | 200 | – | 2048 | 1067 | 216 | – |
| FasterVQ | 4096 | 4096 | 162 | – | 4096 | 4096 | 38 | – |

results show that previous VQ-based methods exhibit extremely poor reconstruction performance, which implies that the VLA backbone is effectively learning a biased supervision signal. Under the same amount of training data, FASTer(S) significantly outperforms Fast. Moreover, even when Fast+ is trained with substantially more data than FASTer(L), FASTer(L) still surpasses Fast+ in reconstruction quality.

## A.6 Use of LLM

We utilized LLMs as a writing assistance tool during the preparation of this manuscript. The use of LLMs was strictly limited to polishing the text, which included improving grammar, refining sentence structure, and enhancing overall clarity and readability. The core research concepts, methodologies, and conclusions were developed entirely by the authors.

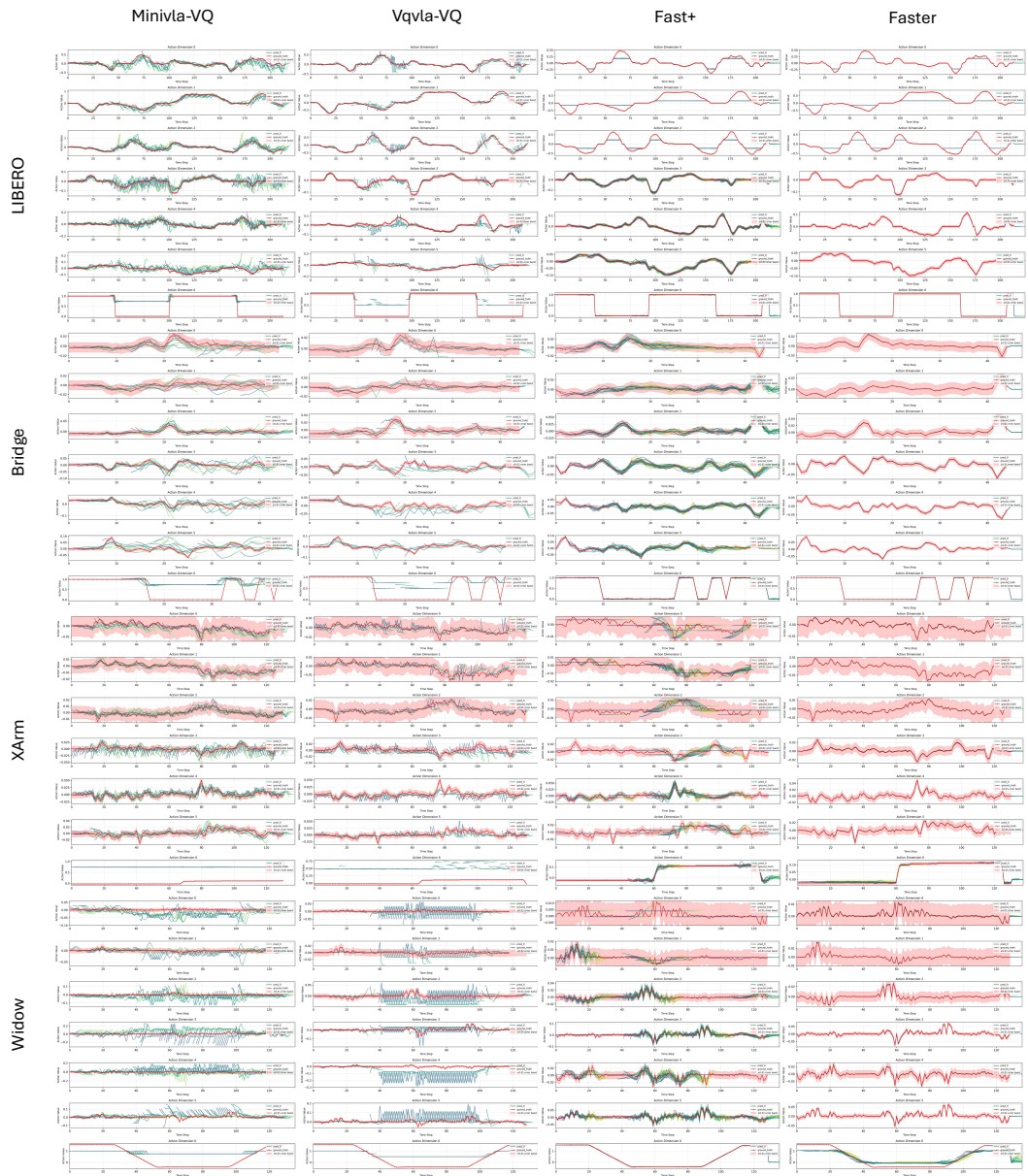

Figure 14: Visualization of action reconstruction results from MiniVLA, VQ-VLA, Fast+, and FasterVQ on representative trajectories from the Libero, Bridge, XArm, and Widow datasets.

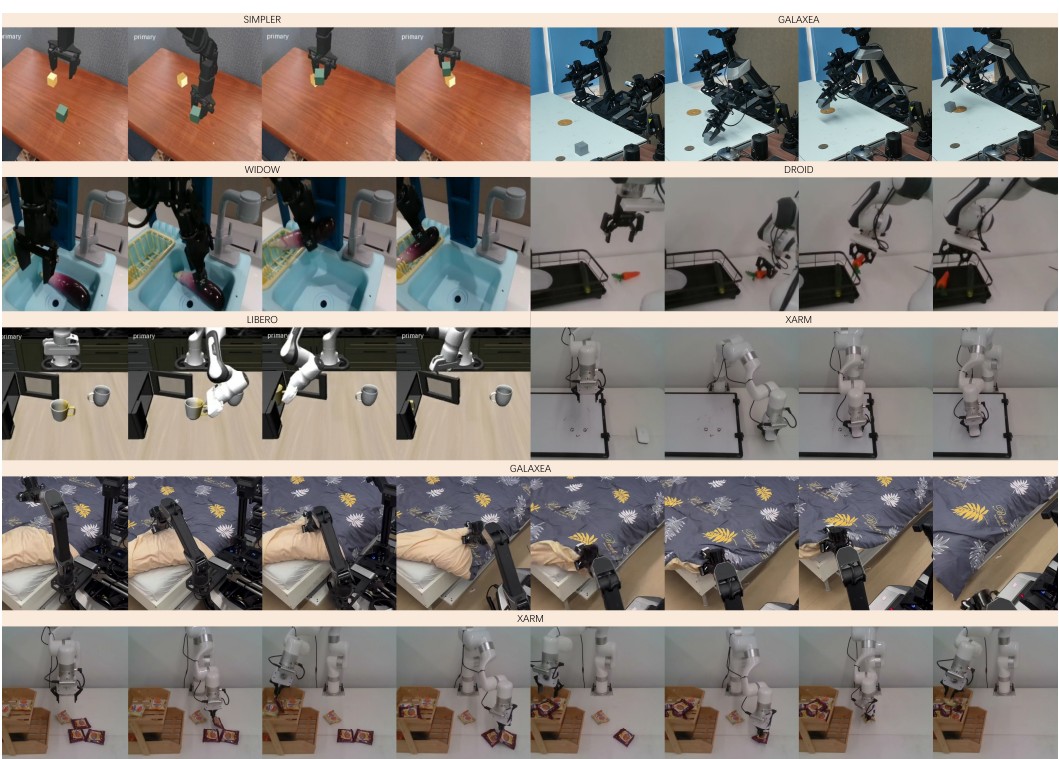

Figure 15: Visualization of our evaluation tasks.

