# OpenReview forum: "FASTer: Toward Powerful and Efficient Autoregressive Vision–Language–Action Models with Learnable Action Tokenizer and Block-wise Decoding"
_ICLR.cc/2026/Conference — ICLR 2026 Poster_

### Official Review · Reviewer_uL4Q · 2025-10-23

**Soundness:** 3
**Presentation:** 3
**Contribution:** 3
**Rating:** 6
**Confidence:** 4

**Summary:**

The paper introduces FASTer, an approach for discrete-token action generation for VLA policies:

- **FASTerVQ (tokenizer)**: A transformer autoencoder with residual VQ (RVQ) over an action patchifier. Training uses a dual-domain reconstruction loss (temporal L1 + DCT-domain L1) to preserve both low and high-frequency motion. The result is a compact, balanced code usage with improved reconstruction.

- **FASTerVLA (policy)**: An autoregressive (AR) policy that (i) adds an action-expert head, (ii) uses RoPE spacing augmentation to avoid positional overfit on fixed horizons, and (iii) employs Block-wise Autoregressive (BAR) decoding with a coarse-to-fine order under a block-causal mask. The paper additionally sketches a “real-time chunking (RTC)” regime for asynchronous control via stochastic training and lightweight logit ensembling.

Across simulators and real-robot setups, the method reports higher success rates and lower inference latency than prior AR tokenizers (e.g., FAST/FAST+), with competitive results to continuous flow/diffusion policies in some regimes. Ablations examine tokenizer design (codebook sizes, RVQ depth, losses) and decoding choices, and provide analysis on vocabulary utilization and reconstruction-error vs compression.

**Strengths:**

- While built from known parts (transformer RVQ with time+DCT loss, block-wise causal masks for AR, coarse-to-fine action decoding), the specific combination is novel for VLA setting and practically impactful for AR control policies.
- Extensive tokenizer analysis (codebook utilization, reconstruction vs compression rates) and sensible ablations on RVQ/codebook design choices. Reported latency reductions are compelling for real-time control discrete AR policies.
- Method components are described clearly. Design choices are well-motivated (e.g., position overfitting on fixed horizon actions, the coarse-to-fine decode order is intuitive and naturally follows RVQ) accompanied by real world and simulation benchmarks supporting the method.
- Improving throughput/latency without giving up success is valuable in robotics. The work should be useful to groups deploying AR VLAs where continuous policies are costly to train.

**Weaknesses:**

It is known that diffusion/flow based are slower to train compared to autoregressive models, the paper should report compute comparison between baselines to accurately place the gains in context. I still believe given sufficient compute continuous diffusion/flow policies would outperform discrete AR policies. AR action heads trained with cross-entropy are brittle for real world control, exhibiting unpredictable bin prediction switches with little observation noise. CE loss is non-metric, it only cares about the probability on the ground-truth bin, no matter how the remaining probability mass is distributed among nearby vs far bins. Continuous policies avoid this by design as they directly penalize errors proportional to deviation. How much does the FASTerVQ tokenization mitigate this?

**Questions:**

- How does Block AR differ from similar approaches used in discrete diffusion literature (see Block Diffusion [1])? It would be good to position this paper appropriately and add references.
- It would be good to have comparisons against $\\pi$-0.5 which is an improved version of $\\pi$-0 with better generalization.
- Was a single policy trained for all LIBERO task suite or each LIBERO suite had a separate policy? OpenVLA-OFT attains higher success rate when using a single policy setting, so the paper should clarify here and update the results accordingly. OpenVLA-OFT also has a FiLM variant which improves language grounding. Please take a look at Appendix in [2] for details and update accordingly.
-  There seems to be some discrepancy in reported latencies for different models and OpenVLA-OFT paper Table III, some clarification here would be helpful. also the main paper should include inference time analysis, detailing out inference latency, action chunk size used and max control frequency supported.
- Could you provide some insight on Figure 5 (c)? Why compression ratio is so different for different dataset?


[1] Block Diffusion: Interpolating Between Autoregressive and Diffusion Language Models

[2] Fine-Tuning Vision-Language-Action Models: Optimizing Speed and Success

---

> ### Author Response · Authors · 2025-11-26
> **Response part1**
>
> Q1. Compute fairness
> > It is known that diffusion/flow based are slower to train compared to autoregressive models, the paper should report compute comparison between baselines to accurately place the gains in context.
>
> Across all experiments, we adhere to the official π₀ fine‐tuning procedure in LIBERO and follow the established diffusion-based VLA training routine used in real-world settings. Under this protocol, the diffusion model is fully trained before comparison, ensuring that differences in performance are not due to inadequate training of any component. Prior observations from FAST[1] indicate that AR models tend to converge more rapidly and reach stronger optima, which further motivates the need for careful protocol alignment.
>
> More concretely, our π₀ baseline is trained with the same configuration as FASTerVLA: a batch size of 32 on 8 H20 GPUs, and an identical model size. We have also updated the appendix (Section Detailed Evaluation Setups) to report the compute used by the remaining benchmarks. All methods operate under a uniform training setup so that the comparison reflects model design rather than differences in computational budget.
>
> The table below shown the performance of FASTerVLA and pi0 models as a function of training steps on the Bridge benchmarks. Both models are trained for eight epochs (33 366 steps). We intentionally extend the training schedule, since in our setting the pi0 model often reaches its best performance around four epochs. The AR model attains a stronger solution earlier and ultimately achieves higher performance, while the pi0 model shows a noticeable degradation after its optimal step. In addition, we observe that prolonging AR training further improves its performance in more complex real-robot experiments (whole-body manipulation), whereas the diffusion model does not seem to benefit in a meaningful way from additional training.
> [1] FAST: Efficient Action Tokenization for  Vision-Language-Action Models
>
> | Training step | pi0 Performance (%) | faster Performance (%) |
> |----------------|----------------------|--------------------------|
> | 5000           | 35.6                 | 50.5                     |
> | 10000          | 59                   | 70                       |
> | 16683          | 65                   | 85.6                     |
> | 25000          | 66                   | 88.1                     |
> | 33366          | 62.5                 | 88                       |

---

> > ### Author Response · Authors · 2025-11-26
> > **Response part2**
> >
> > Q2. question about using CE loss as objective.
> > > I still believe given sufficient compute continuous diffusion/flow policies would outperform discrete AR policies. AR action heads trained with cross-entropy are brittle for real world control, exhibiting unpredictable bin prediction switches with little observation noise.
> > > CE loss is non-metric, it only cares about the probability on the ground-truth bin, no matter how the remaining probability mass is distributed among nearby vs far bins. Continuous policies avoid this by design as they directly penalize errors proportional to deviation. How much does the FASTerVQ tokenization mitigate this?
> >
> > We appreciate the reviewer’s concern regarding the non-metric nature of CE and its potential brittleness. We agree that such issues may arise when autoregressive action heads are trained from scratch in low-data regimes. However, in our setting with extensive pretraining and sufficient data, discrete autoregressive VLA models, especially FASTer, can perform well. Our answer comes from two complementary perspectives.
> >
> > First, FASTerVQ implicitly encodes much of the geometric structure of the action space. The VQ stage is trained with an L1 reconstruction loss, which encourages the encoder to embed similar actions into nearby regions of a continuous latent space. The residual vector quantization produces coarse-to-fine action codes whose perturbations remain locally constrained. As a result, mispredictions remain bounded in action space, and the decoded actions drift only slightly under small observation noise. This behavior is also illustrated in the middle of Figure 1, where even a partially predicted sequence of FASTerVQ codes can already reconstruct an action trajectory that stays very close to the ground truth. In this sense, the local continuity of actions is encoded by FASTerVQ.
> >
> > Second, in the VLA training stage, we deliberately avoid imposing additional inductive bias. The model must learn not only the local continuity of actions, but also the distribution of actions across different visual and linguistic contexts. Two actions that are close in geometric terms may play fundamentally different functional roles depending on the observation. Imposing metric penalties on the action space risks conflating these distinctions or encouraging unintended averaging, particularly in dexterous imitation settings where each expert trajectory often represents a single viable mode. CE preserves the demonstration-centered precision required by these tasks, allowing the model to learn contextual relations directly from paired observations and instructions rather than from an externally imposed notion of proximity.
> >
> > Taken together, FASTerVQ encodes the local geometry of the action space, while the CE objective allows the VLA to learn the context-dependent action distribution without assuming a global metric structure that may not align with expert demonstrations. This combination yields a balanced trade-off: geometric robustness inherited from the tokenization stage and contextual fidelity preserved by CE-based autoregressive modeling.
> >
> >
> > Q3.Compare with block diffusion
> > > How does Block AR differ from similar approaches used in discrete diffusion literature (see Block Diffusion [1])? It would be good to position this paper appropriately and add references.
> >
> > Thank you for the suggestion. We have added Block Diffusion to the related work section. While Block AR shares the idea of preserving causal dependencies across blocks and allowing fully visible parallel decoding within each block, the underlying mechanisms differ substantially. Block Diffusion requires a full denoising process for every block, which introduces multiple NFEs per step. For example, for a sequence of length 2048 with a block size of 4, it requires approximately 2k forward passes. In contrast, our block-wise autoregressive formulation only needs 2048/4 = 512 forward passes. This efficiency stems from the assumption that action-dimension tokens are independent within a block and do not require conditional generation. Our method retains a purely autoregressive objective, whereas Block Diffusion relies on a noise-adding and denoising procedure.

---

> > > ### Author Response · Authors · 2025-11-26
> > > **Response part3**
> > >
> > > Q4. OpenVLA-OFT issues
> > > > - Was a single policy trained for all LIBERO task suite or each LIBERO suite had a separate policy? OpenVLA-OFT attains a higher success rate when using a single policy setting, so the paper should clarify here and update the results accordingly. OpenVLA-OFT also has a FiLM variant which improves language grounding. Please take a look at Appendix in [2] for details and update accordingly.
> > >
> > > Thank you for pointing it out. We will update the OpenVLA-OFT results. We have already incorporated its performance into our manuscript, and indeed we reported the OpenVLA-OFT scores using the discrete diffusion policy [1]. In the updated Table 1, we use the strongest OpenVLA-OFT result, namely the “one policy per suite” setting. In contrast, our LIBERO experiments use a single policy jointly trained across all LIBERO task suites.
> > >
> > > [1] Discrete Diffusion VLA: Bringing Discrete Diffusion to Action Decoding in Vision-Language-Action Policies
> > >
> > > Q5. inference analysis
> > > > There seems to be some discrepancy in reported latencies for different models and OpenVLA-OFT paper Table III, some clarification here would be helpful.
> > > > Also, the main paper should include inference time analysis, detailing out inference latency, action chunk size used and max control frequency supported.
> > >
> > > We conducted our tests using our own reproduction of pi0, so the forward inference is slightly slower compared with implementations using FlashAttention or the optimized pi0 version. Therefore, there are some discrepancies between our results and those reported in openVLA-OFT. To provide a more detailed view of the inference time, we include a breakdown below. All tests in this table were run in a PyTorch-based codebase on an RTX 5090. “Single” refers to the single-arm setting with an input chunk size of 20 and two camera views. WBC refers to a 21-DoF whole-body control setting with a chunk size of 32 and three camera views. We have also updated our main paper with these results.
> > >
> > > | **model part**              | **Time (Single)** | **Time (WBC)** |
> > > |-----------------------------|-------------------|----------------|
> > > | image encoders              | 16 ms             | 23 ms          |
> > > | observation forward pass    | 72 ms             | 105 ms         |
> > > | AR forward pass             | 6.4 × 21 ms       | –              |
> > > | BAR forward pass            | 7.4 ms × 3        | 8.51 ms × 12   |
> > > | FASTerVQ detokenization     | 2.7 ms            | 7 ms           |
> > > | **total inference**         | **112 ms**        | **237 ms**     |
> > >
> > > We updated the control frequency of FASTer in Figure 1 and in the table below. The control frequency is obtained by dividing the episode execution time by the number of controller steps, so the model’s inference delay is naturally embedded in the resulting frequency.
> > >
> > > | System         | Xarm | Widow | Libero | Droid | VLA Bench | Simpler-Widow | R1 Lite  Binmanual | R1 Lite WBC |
> > > |----------------|------|-------|--------|-------|-----------|----------------|---------|-------------|
> > > | Chunk Size     | 10   | 10    | 20     | 10    | 10        | 10             | 32      | 32        |
> > > | Control Freq   | 11.9 | 6.3   | 7.1    | 9.9   | 5.2       | 6.6            | 6.3     | 4.5       |          |
> > >
> > >
> > > Q6. Compare with pi0.5
> > > > It would be good to have comparisons against -0.5 which is an improved version of -0 with better generalization.
> > >
> > > We additionally evaluate Pi05 on LIBERO, LIBERO (Scratch), VLABench, and XArm Suite; the results are summarized in the following Table. To ensure apples-to-apples comparison with the rest of our paper—which is implemented in PyTorch—our supplementary Pi05 experiments also use the official PyTorch implementation.
> > > Our experiments indicate that the current open-source implementation of Pi05 does not outperform Pi0/PiFast to the extent reported by the original paper, though its performance remains comparable.
> > >
> > > | Dataset / Method     | pi0  | pifast | pi05 | faster |
> > > |----------------------|------|--------|------|--------|
> > > | LIBERO               | 94.2 | 94.2   | 96.8 | 97.9   |
> > > | LIBERO (scratch)     | 89.2 | 88.55  | 81.2 | 94.8   |
> > > | VLBench (ID)         | 14.9 | 10.6   | 13.4 | 16.2   |
> > > | Xarm Suite (Real)    | 31.33| 29.92  | 21.4 | 46.67  |

---

> > > > ### Author Response · Authors · 2025-11-26
> > > > **Response part 4/4**
> > > >
> > > > Q7 Insight of Figure5
> > > > > Could you provide some insight on Figure 5 (c)?
> > > > > Why compression ratio is so different for different dataset?
> > > >
> > > > **Insights of Figure 5(c)**
> > > >
> > > > In the original Figure 5(c), we present the compression rates of different action tokenizers across multiple datasets. Our intention was to highlight that although the previous VQ-Base tokenizer achieved extremely high compression by using very few tokens for long action sequences, it incurred severe reconstruction errors. In the revision, we updated this figure to Figure 6, and additionally report a simple composite metric, VRR × CR, which jointly reflects reconstruction fidelity and compression efficiency. This metric provides a more intuitive view of how FASTer simultaneously balances compactness and fidelity, highlighting its advantage over prior action tokenizers. Moreover, we include the action chunk sizes for each dataset in Figure 6. We can observe the trend that FASTer achieves consistently high compression quality and demonstrates superior performance on datasets characterized by high-frequency action sequences or long-horizon trajectories.
> > > >
> > > > **The Difference Compression Ratio between Datasets**
> > > >
> > > > The observed differences in compression ratios across datasets primarily stem from differences in action frequency. Following the action-chunk modeling strategy in [1, 2], the sequence length within a fixed temporal window (e.g., one second) can vary substantially across datasets.
> > > >
> > > > For example, when FASTer performs compression along the temporal horizon dimension H, it maps the horizon dimension of each action dim into N tokens. As a result, the compression ratio of FASTer is approximately H / N; in our experiments, N is typically set to 3. Consequently, the compression ratio naturally varies with the horizon length of the action chunk.
> > > >
> > > > In comparison, the FAST tokenizer applies BPE-style compression over flattened action sequences, meaning its compression ratio also varies with the underlying action sequence length. The original FAST paper [2] similarly reports higher compression ratios for higher-frequency action sequences. In our experiments, we observe the same trend; however, FASTer consistently achieves stronger compression efficiency on high-frequency action sequences than FAST under comparable conditions.
> > > >
> > > > [1] Learning Fine-Grained Bimanual Manipulation with Low-Cost Hardware.
> > > >
> > > > [2] FAST: Efficient Action Tokenization for Vision-Language-Action Models

---

### Official Review · Reviewer_DL7K · 2025-10-26

**Soundness:** 4
**Presentation:** 3
**Contribution:** 3
**Rating:** 8
**Confidence:** 4

**Summary:**

This paper studies the action tokenization problem, which is valuable for performance and efficiency of autoregression-based policy. The authors propose FASTer, an improved version upon FAST, by incorporating learnable vector quantization (instead of fixed procedure) to improve the compactness and fidelity of action tokenization.

**Strengths:**

- The paper studies an important problem that carries strong practical applications.
- The proposed method makes sense, is simple yet effective.
- The proposed method is extensively evaluated with different metrics, in diverse robot environments, including real-world.

**Weaknesses:**

- The texts in many figures are quite small and not easy to read especially when printed, particularly in Fig. 2 and Fig. 3.

- Page 4 details the action tokenization process and components. I believe it would be easier for readers if there is an algorithm block that outlines this procedure, like FAST does. Currently, it takes some effort to gather everything from text.

- [1] is a recent work that incorporates autoregressive action chunk generation without an explicit action tokenizer, and its proposed technique is very relevant to the block-wise autoregression. The authors are recommended to include this in the related work.   [1] Autoregressive Action Sequence Learning for Robotic Manipulation (RAL 2025)

**Questions:**

- How does it work with diverse robots and action types? If I train a single FASTer tokenizer for multiple robots, does this mean (1) a slower encoding/decoding process, (2) a less compact coding?

- Fig7 shows FASTer underperforms than Pi0 in in-distribution tasks. I understand the emphasis is OOD performance in that figure, but why is that?

- Does the space augmentation only change position embedding, or actually change the sequence values? If the latter, is there an example of the space augmentation?

- Regarding the block-wise autoregression, I am very interested in the step size (block size) in the autoregression. In the paper, the step size = B (size of one codeblock). But what would the performance possibly be if the block size is 1 or N, or other number? Does a irregular granularity will affect autoregression performance?

- Is it possible to provide some failure cases and related analysis?

---

> ### Author Response · Authors · 2025-11-26
> **Response part1**
>
> Q1.Small figure text.
> > The texts in many figures are quite small and not easy to read especially when printed, particularly in Fig. 2 and Fig. 3.
>
> Thanks for the suggestion. We have enlarged the text size in Figures 1, 2, and 3.
>
> Q2. Add a algorithm block that outlines this procedure
>
> We have add a algorithm block in the revised paper.
>
> Q3. Add reference
>
> We have included this work in the revised related work section. In addition, we added a dedicated subsection that discusses block-wise or chunk-level generation, in which we position this recent approach within the broader landscape and clarify its relationship to our method.
>
> Q4. Questions of Universal tokenization
> > How does it work with diverse robots and action types? If I train a single FASTer tokenizer for multiple robots, does this mean (1) a slower encoding/decoding process, (2) a less compact coding?
>
> Thank you for raising this important question regarding the use of FASTer across diverse robots and action types. This represents a valuable research direction centered on data scaling. We address the reviewer’s two concerns below.
>
> **(1) Does training a single FASTer tokenizer for multiple robots lead to slower encoding/decoding?**
>
> The answer is no. The encoding and decoding speed of FASTer is determined solely by the tokenizer model size, rather than by the diversity or amount of action data it is trained on. In our current design, FASTer(S) and FASTer(L) differ only in the scale of training data—their architecture and model capacity remain identical—so their runtime latency is the same around 3ms.
>
> To further verify that action diversity does not slow down FASTer, we trained an additional variant, **FASTer(XL)**, using **3× more heterogeneous data**, including joint positions, delta end-effector motions, relative-chunk trajectories, and Droid-style velocity commands across multiple embodiments. The updated results in Figure 5 show that increasing the diversity and quantity of training data does **not** incur additional encoding/decoding cost, confirming that runtime efficiency is unaffected.
>
> Finally, regarding model-size scaling, we may have a disscusion here: while our current exploration focuses on data scaling (increasing training coverage while keeping model size fixed), we agree that **scaling the tokenizer model itself**—similar to recent trends in speech representation learning—is a promising direction. We plan to explore model-capacity scaling in future work, but our current goal is to maintain a tokenizer that is **small, fast, and suitable for real-time robotic inference**. And the results show that even a relatively small tokenizer is already capable of accommodating this amount of diverse action data effectively.
>
> **(2) Does supporting multiple robots make the representation less compact?**
>
> As discussed above with FASTer(XL), our empirical results show that increasing the amount and diversity of training data does not reduce the compression rate. On the contrary, under the same compression ratio, the tokenizer achieves better reconstruction performance.
>
> Unlike Fast, whose token sequence length varies with trajectory complexity, FASTer is designed around the **2D structural properties of robot actions** and uses a **fixed number of tokens** to compress the horizon dimension. This means that the compression efficiency depends only on the residual hyperparameter and the action-chunk horizon length. We further extended the horizon length while keeping the rule of “three tokens per action dimension,” and FASTer continued to perform well.
>
> These experiments collectively demonstrate that **scaling the data coverage—both in quantity and diversity—substantially improves the performance of the action tokenizer**, even without increasing model size. In other words, before scaling the autoregressive model, scaling the action tokenizer itself already yields significant benefits.

---

> > ### Author Response · Authors · 2025-11-26
> > **Response part2**
> >
> > Q5. question about figure7
> > > Fig7 shows FASTer underperforms more than Pi0 in in-distribution tasks. I understand the emphasis is OOD performance in that figure, but why is that?
> >
> > There are two main reasons that FASTer appears to underperform Pi0 on in-distribution tasks in the earlier version of original Figure 7:
> >
> > 1. The earlier experiments used a smaller FASTerVQ tokenizer (FASTerVQ-S).
> >
> > In the initial setup, our tokenizer was trained only on the VLABench dataset (“FASTerVQ-S”). We later found that training the tokenizer on larger and more diverse data leads to **significantly better reconstruction quality** and a **more balanced token distribution**. After replacing FASTerVQ-S with the improved **FASTerVQ-L**, we re-ran the experiments and observed that the overall task success rate **increased by about 10% relative**, including on in-distribution tasks.
> >
> > 2. VLABench tasks exhibit high episode-level variance.
> >
> > Unlike Libero, the VLABench dataset has large distribution shifts between episodes even within the same task. Thus, multiple runs under the identical experimental setup naturally yield some variance in the resulting performance.
> > We therefore evaluated **three different seeds**, aggregated the results, and updated the average success rate and its std in the new **Figure 9**.
> >
> > 3. Initialization Bias
> >
> > Additionally, **the initialization scheme is not entirely fair to FASTer**. FASTer-VLA is initialized from PiFast, whereas Pi0 uses the official pretrained checkpoint, which was trained on 10,000 hours of robotics data. This mismatch means FASTer starts from a comparatively less optimized initialization, which can disadvantage its performance in some settings.
> >
> > Q6.Explain the space augmentation
> > > Does the space augmentation only change position embedding, or actually change the sequence values? If the latter, is there an example of the space augmentation?
> >
> > In RoPE, the position ID determines the rotation angle for every frequency component. We only modify these position IDs, and the token embeddings themselves remain unchanged. This means the augmentation alters the effective geometric distance between tokens in the RoPE coordinate system, without changing the RoPE computation rule. An intuitive way to view this is that the position IDs directly decide how far apart tokens are in the rotational embedding space; spacing augmentation simply perturbs these distances.
> >
> > For example, a vanilla sequence may use position IDs: 1, 2, 3, 4, 5, 6, 7.   With spacing augmentation, a training sequence might instead be encoded as: 1, 3, 6, 7, 9, 11, 14  or  2, 4, 5, 8, 10, 12, 13,  depending on the sampled jitter. During inference, we revert to a fixed stride such as  2, 4, 6, 8, 10, 12, 14.
> >
> > Q7. BAR ablation study
> > > Regarding the block-wise autoregression, I am very interested in the step size (block size) in the autoregression. In the paper, the step size = B (size of one codeblock). But what would the performance possibly be if the block size is 1 or N, or other number? Does a irregular granularity will affect autoregression performance?
> >
> > Our block size is determined by the number of codes along the action-dimension axis, and the block size is always an integer multiple of the number of action-dimension codes. When the block size equals one, it reduces to a standard autoregressive model. We evaluated different block sizes within the BAR framework using the same FASTerVQ model. The length of our action code is 42, and the detailed results are shown in the table below and figure 12 in revised paper. When the block size is close to the action-dimension code length, the performance differences are minor; when the block size deviates substantially from the action-dimension code length, the performance degrades. The best performance is achieved when the block size matches the action-dimension code length. These observations are consistent with our initial design intuition. BAR is inherently coupled with FASTerVQ, because FASTerVQ preserves the structure of action chunks, allowing us to exploit this structural property for block-wise generation.
> >
> > | Block size | Libero performance |
> > |------------|--------------------|
> > | 1          | 95.5               |
> > | 2          | 95.9               |
> > | 3          | 96.65              |
> > | 6          | 96.85              |
> > | 7          | 97.7               |
> > | 14         | 95.75              |
> > | 16         | 95.95              |
> > | 21         | 93.85              |
> > | 24         | 93.70              |

---

> > > ### Author Response · Authors · 2025-11-26
> > > **Response part 3/3**
> > >
> > > Q8. Is it possible to provide some failure cases and related analysis?
> > >
> > > In our evaluation, we observed two primary types of failure cases. First, the model may occasionally decode an incorrect sequence length. This issue can be addressed by enforcing the output length during decoding. Second, due to the imbalance in the dataset, the model sometimes tends to overproduce the no-operation action. We alleviate this tendency by applying nucleus sampling, which is commonly used in LLM decoding to reduce such bias. Moreover, because autoregressive models fit the data distribution more tightly, they are more sensitive to distributional skew; when the dataset contains a large proportion of no-operation actions, the model can become inclined to generate extended periods of inactivity.
> > >
> > > In addition, because we use CE loss during training without any inductive bias, the model may learn subtle action patterns from expert demonstrations when the data distribution is skewed. For example, in another dataset that we collected, the operator tended to slightly rotate the wrist of the non-active hand whenever the other hand was moving. The model consequently learned this pattern and reproduced the same minor wrist rotation.

---

### Official Review · Reviewer_8Nbo · 2025-10-27

**Soundness:** 3
**Presentation:** 3
**Contribution:** 3
**Rating:** 4
**Confidence:** 2

**Summary:**

This paper is about making (auto-regressive) vision language action models more useful in contexts like robotics where fast inference and precise reconstruction are important for overall system performance and inference time impacts which action frequency tasks can have. The main focus is on the action tokenization part. The approach designs action tokens (action representation) by performing residual vector quantization and  trains reconstructing action sequences in the time and frequency domain (loss functions). This is supposed to result in compact and highly compressed action sequences. The inference approach infers multiple tokens in parallel by block-wise decoding. The paper also establishes a larger benchmark, collecting several real robots and simulations.

Importantly, this approach is supposed to surpass non-auto-regressive models in inference speed and reconstruction accuracy.

**Strengths:**

- The paper is nicely written and pedagogical, especially the introduction section and the related work section.

- The general idea of patchifying the continuous actions seems very reasonable.

- RVQ is generally very suitable for this problem.

- Adding the DCT to the step-wise reconstruction loss makes sense for capturing the overall trend as long as the action sequences behave with trends within horizon H.

- The paper contains a large set of experiments and evaluations which provide insight in the performance and tradeoffs of the proposed method.

**Weaknesses:**

- The reusability in some figures is low due to the choice of colors, lack in contrast, and lack of clear captions. Page 7 is a good example for this issue. Red-green contrasts should be avoided in general.

- There are a few types in the manuscript, repeated words, and wrong references (e.g. to Fig. 5 on page 7).

**Questions:**

- When introducing the VRR measure for evaluation, the authors write that a parameters is set "corresponding to an allowed error of approximately 1cm". It is not clear what this means. Is this 1 cm Euclidean error at the end-effector pose (in case of manipulation)? For many robotic tasks, 1 cm error is substantial and means that the execution of the task will fail. The authors might want to elaborate on the VRR. What happens if the threshold is set of 0.5 cm or 1 mm?

- The related work section makes little effort to contrast the proposed method with the state of the art. While the experiments speak for themselves in terms of quantitative results, it would be an improvement if the authors could also argue the improvement over the state of the art conceptually.

---

> ### Author Response · Authors · 2025-11-26
> **Response**
>
> Q1. Issues of Figures
> > - The reusability in some figures is low due to the choice of colors, lack in contrast, and lack of clear captions. Page 7 is a good example for this issue. Red-green contrasts should be avoided in general.
>
> Thank you for the helpful suggestion. We have revised the figures and their corresponding captions to improve readability and visual clarity. Specifically, for the original Figure 5(a)(b), we moved the detailed heatmap results to Figure 13 in Appendix A.5 and replaced the original red–green color scheme with a monochromatic green-scale heatmap. This figure reports the VRR of different tokenizers across multiple datasets and different σ values in a more visually coherent manner.
>
> In the revised manuscript, we also updated Figure 5 and Figure 6 along with their captions. Figure 5 now presents the average VRR performance under different σ thresholds using line plots, which makes the comparison between tokenizers more intuitive. In addition, Figure 6 is updated to provide a clearer visualization of the trade-off between reconstruction fidelity and compression efficiency, highlighting that FASTer achieves a more favorable balance than prior approaches.
>
> Additionally, we refined several visual details throughout the paper, including increasing font sizes in Figures 1–3 and standardizing line widths and shadow styles in the experimental bar charts. We hope these improvements enhance the overall readability and presentation quality of the paper.
>
> Q2. Typos
> > - There are a few types in the manuscript, repeated words, and wrong references (e.g. to Fig. 5 on page 7).
>
> Thank you for pointing this out. We will carefully revise the manuscript to eliminate minor textual issues, including typos, repeated phrases, and reference inconsistencies. For example, we have corrected the spelling errors in “bridge” and “compression” in the original Figure 5.
>
> Q3. Issues of Related works
> > The related work section makes little effort to contrast the proposed method with the state of the art. While the experiments speak for themselves in terms of quantitative results, it would be an improvement if the authors could also argue the improvement over the state of the art conceptually.
>
> We have revised the related work section to provide a more explicit conceptual comparison with the state of the art. In addition to incorporating this analysis, we have added a new subsection that discusses recent progress on block-wise generation and clarifies how our approach differs from and improves upon prior methods.
>
> Q4.Issues of VRR
> > When introducing the VRR measure for evaluation, the authors write that a parameters is set "corresponding to an allowed error of approximately 1cm". It is not clear what this means. Is this 1 cm Euclidean error at the end-effector pose (in case of manipulation)? For many robotic tasks, 1 cm error is substantial and means that the execution of the task will fail. The authors might want to elaborate on the VRR. What happens if the threshold is set of 0.5 cm or 1 mm?
>
> Thank you for pointing out the ambiguity.  In the revision, we explicitly define σ as the per-dimension reconstruction tolerance used by VRR: for **robot end-effector translation**, σ corresponds to the Euclidean distance error in meters; for **robot end-effector rotation** and **joint positions**, σ represents an angular error measured in radians. Empirically, thresholds near σ = 1e-2 lead to noticeable performance drops; this is **particularly pronounced for end-effector translation**, since ~1 cm position error is often task-critical.
>
> Considering both information density and visual clarity, we have redrawn **Figure 5** and now explicitly illustrate the VRR achieved by different tokenizer methods under multiple σ scales. As shown in Figure 5, across the σ range from **1e-2 to 5e-4**, FASTer consistently outperforms Fast, even when trained on equal or fewer samples than Fast (comparable data budgets). At σ=**1e-4**, the reconstruction error becomes almost negligible for manipulation tasks as we found that motor noise is at the level of 1e-4 , and FASTer and Fast exhibit comparable performance, whereas prior VQ-based methods yield VRR values close to zero at this scale. We provide the detailed heatmaps of evaluation results across multiple datasets in Figure 13 in Appendix A.5, with a more visually coherent and harmonious color scheme.

---

### Official Review · Reviewer_8e91 · 2025-10-30

**Soundness:** 4
**Presentation:** 3
**Contribution:** 3
**Rating:** 8
**Confidence:** 3

**Summary:**

The paper proposes a new learnable tokenizer scheme based on residual vector quantization (RVQ) for robotic manipulation datasets. A custom action patchifier is used to group timesteps and physically similar action dimensions together which is subsequently fed into an RVQ model learning a coarse-to-fine representation of action signals. The tokenizer is trained on a combination of RVQ losses alongside local and global action reconstruction objectives. Furthermore, a custom VLA architecture utilizing the new tokenizer is presented which adds block-wise autoregressive prediction of multiple tokens with a custom decoding order to speed up inference.

**Strengths:**

-	Overall, the paper is well written and clearly motivates the drawbacks of previous tokenizer designs in VLA models. It encompasses the most relevant previous work and gives a clear outlook of which open problems in the field it tackles.
-	The proposed tokenizer (FASTer) for VLA models addresses relevant challenges of action tokenization, such as achieving good reconstruction quality with high compression rates. The choice of RVQ for this task seems adequate and timely: Learnable tokenizers have shown strong initial results in robotics but rely on partially manual design for converting continuous signals into discrete codes.
-	The use of a custom decoding order in combination with the RVQ tokenizer is well thought out, resembling the strengths of diffusion-based action prediction heads (going from low frequency to high frequency reconstruction) without the need for diffusion training.
-	An extensive list of experiments is conducted, comparing FASTer VLA with relevant baselines such as pi0 using the FAST tokenizer on a wide range of benchmarks. Aside from inference speedups, FASTer VLA achieves broad success rate improvements across the board with only VLABench lacking behind, suggesting strong generality of the method.
-	The performance of the FASTer tokenizer itself is also analyzed properly, showing good scalability and generalization to out of distribution data.

**Weaknesses:**

-	Aside from the LIBERO experiments in Table 1, most results seem to combine FASTer VLAs blockwise autoregressive decoding (BAR) with the FASTer tokenizer, making it difficult to judge the impact of the tokenizer itself on success rates. Optimally, a tokenizer should be mostly agnostic to the choice of architecture. This is partially shown in Figure 6 with different VLM backbones but still might rely on custom decoding schemes (see first question).
-	The encoding times of the FASTer tokenizer seem high (150ms on LIBERO) compared to the runtime of the base pi0 model. To the best of my understanding, the paper does not isolate the tokenizer runtime as a standalone metric in the inference speed results.

**Questions:**

-	It seems that BAR decoding in combination with FASTer leads to significantly higher success rates than just using FASTer. Could we expect similar gains when using BAR in combination with other tokenizers? I would be curious to see the performance when using the FAST tokenizer with BAR to fully determine the benefits of FASTer versus FAST.
-	Similar to the previous question, I  would be curious how much using BAR decoding offsets the potentially slow tokenizer.

---

> ### Author Response · Authors · 2025-11-26
> **Response part1**
>
> Q1: question about block-wise autoregression
> >Aside from the LIBERO experiments in Table 1, most results seem to combine FASTer VLAs blockwise autoregressive decoding (BAR) with the FASTer tokenizer, making it difficult to judge the impact of the tokenizer itself on success rates. Optimally, a tokenizer should be mostly agnostic to the choice of architecture.
> This is partially shown in Figure 6 with different VLM backbones but still might rely on custom decoding schemes (see first question).
>
> Indeed, we only compared the effect of adding BAR versus not adding BAR for the LIBERO and BRIDGE settings. We agree with your point that a tokenizer should be mostly agnostic to the choice of architecture. Therefore, we additionally included comparisons of BAR and non-BAR variants under different model structures, namely Qwen and InternVL. Results are shown blow and Figure7 in our revised paper. From these results, BAR provides some improvement across architectures, but the primary gains still come from the VQ design itself.
>
> | Model          | FAST  | FASTer w/o BAR | FASTer |
> |----------------|-------|----------------|--------|
> | Pailigemma2-3B | 93.50 | 94.00          | 94.80  |
> | Qwen2.5-3B     | 91.30 | 95.40          | 95.45  |
> | InternVL3.5-2B | 79.35 | 96.30          | 96.65  |
>
> To further address your questions about BAR, we explored more possibilities: applying BAR to FAST and using more irregular block sizes. We first attempted to apply BAR directly on FAST. Concretely, we padded FAST’s variable-length action codes to the nearest multiple of the block size, and then trained these blocks with teacher forcing. However, even after training for 60k steps (more than twice our normal training length) and trying block sizes from 2 to 7, FAST was unable to generate action codes with the correct length during inference. We believe this is inherently difficult for the model because, first, neighboring FAST tokens have little mutual dependency and the flattening operation introduces abrupt discontinuities, and second, the model would have to learn both variable-length outputs and whether the final block should include padding. This combination appears too challenging.
>
> Our view is that BAR becomes feasible when the action-token length is relatively stable and when correlations exist among tokens. Under these conditions, BAR can be applied directly. We tested different BAR block sizes on the same FASTerVQ with action code length 42. The results are shown in the table and figure 12 in revised paper: as long as the block size is within a reasonable multiple of the action-dimension code length, the performance remains similar. When the block size matches the action-dimension code length, the performance is the best. Empirically, these findings align with our design intuition: BAR is tightly coupled with FASTerVQ, because FASTerVQ preserves the inherent structure of action chunks, which allows us to exploit this structure for block-wise generation.
> | Block size | Libero performance |
> |------------|--------------------|
> | 1          | 95.5               |
> | 2          | 95.9               |
> | 3          | 96.65              |
> | 6          | 96.85              |
> | 7          | 97.7               |
> | 14         | 95.75              |
> | 16         | 95.95              |
> | 21         | 93.85              |
> | 24         | 93.70              |

---

> ### Author Response · Authors · 2025-11-26
> **Response part2**
>
> Q2: Tokenizer runtime
> > The encoding times of the FASTer tokenizer seem high (150ms on LIBERO) compared to the runtime of the base pi0 model. To the best of my understanding, the paper does not isolate the tokenizer runtime as a standalone metric in the inference speed results.
>
> > Similar to the previous question, I would be curious how much using BAR decoding offsets the potentially slow tokenizer.
>
> Thank you for highlighting a point that may confuse readers. In our paper, the term encoding time refers to the VLM’s initial forward pass that processes the image, language instruction, and proprioception inputs. This stage, rather than tokenization itself, is the dominant contributor to inference latency. In practice, the action tokenizer accounts for only a very small portion of the overall runtime.
>
> We have rewritten the “Inference Efficiency of FASTer’’ section to clarify this issue. In addition, we now report the detailed runtime of each inference component and examine how BAR decoding affects the overall latency in the table below and in Table 2 of the revised paper. In the Single setting, the single-arm configuration in the LIBERO environment uses an action chunk size of $20$ with two camera views; in the WBC setting, the $21$-DoF configuration uses an action chunk size of $32$ with three camera views. All measurements were obtained using our PyTorch implementation on an RTX 5090 (our RTX 4090 workstation was unavailable), which leads to slight differences from earlier numbers. the
>
> The results indicate that the primary bottleneck is indeed the observation encoding stage. Both pi0 and our model require approximately 88–127 ms for this step. In contrast, the action tokenizer is relatively lightweight, adding only around 2.7–7 ms. Because BAR generates multiple tokens per forward pass, it substantially reduces latency when the number of required action tokens is large.
>
> | **model part**              | **Time (Single)** | **Time (WBC)** |
> |-----------------------------|-------------------|----------------|
> | image encoders              | 16 ms             | 23 ms          |
> | observation forward pass    | 72 ms             | 105 ms         |
> | AR forward pass             | 6.4 × 21 ms       | –              |
> | BAR forward pass            | 7.4 ms × 3        | 8.51 ms × 12   |
> | FASTerVQ detokenization     | 2.7 ms            | 7 ms           |
> | **total inference**         | **112 ms**        | **237 ms**     |

---

### Author Response · Authors · 2025-11-26
**General Response**

We acknowledge that our submission was made late in the rebuttal window, and the subsequent ICLR leak incident unfortunately prevented reviewers from offering further feedback or score adjustments. We fully understand this situation and sincerely appreciate the AC’s hard work throughout the process under such challenging circumstances. In addition, we thank the reviewers for their thoughtful reading and constructive insights. To facilitate the AC’s assessment, we summarise below the substantive revisions already incorporated into the updated manuscript.

We refined the manuscript by correcting typos, removing repeated words, and improving figure clarity. The related work section has been expanded with clearer conceptual structure. We added a new algorithm block on page 4, redrew several figures with enlarged text for readability, and clarified the definition and motivation behind VRR. Substantial new experiments have been added, including BAR ablations across backbones in figure 7, inference runtime breakdowns in table 2, additional analyses on block-wise autoregression in page 18, space augmentation explanation in page 5, and model training behaviour in page 17-18. Updated OpenVLA-OFT and Pi0.5 results are included in Table 1, together with new ablations in Table 2 and extended plots in Figures 5, 9, and 12–13.

---
## Reviewer 8e91
Concern 1: BAR ablations are insufficient and lack comparisons across backbones.

Concern 2: The reviewer asks for tokenizer decoding runtime and BAR’s effect on overall inference speed.

**Response:**

 We added BAR ablations across multiple backbones and provided a detailed runtime breakdown. These updates appear on page 18, Figure 12, and Table 2 in the revised paper.

---
## Reviewer 8Nbo
Concern 1: Presentation issues, including figure clarity and typos.

Concern 2: Related work lacks conceptual comparison.

Concern 3: VRR requires clearer explanation.


**Response:**

 We carefully rechecked the manuscript, corrected the noted typos, and improved the affected figures. The original Figures 5(a)(b) have been moved to Figure 13 in Appendix A.5 for clearer structure. We added a new subsection in related work to highlight conceptual distinctions. We also clarified the definition and sensitivity of VRR on page 7 and redrew Figure 5 to enhance readability.

---
## Reviewer DL7K
Concern 1: Small text size in Figures 1–3, absence of algorithm block, and a missing citation.

Concern 2: Clarification needed for Figure 7 and space augmentation.

Concern 3: More BAR ablations and FASTer failure cases requested.

**Response:**

 We added an algorithm block on page 4, enlarged the text in relevant figures, and incorporated the suggested citation. We expanded the explanation of space augmentation and clarified the insights behind Figure 7. Additional BAR ablations and common failure cases are now included in both the paper and our response.

---
## Reviewer uL4Q
Concern 1: Fairness of diffusion vs autoregressive comparison, validity of cross-entropy for VLA learning, and distinction between Block AR and block diffusion.

Concern 2: OpenVLA-OFT numbers appear incorrect.

Concern 3: Requests a detailed inference-time breakdown and clarification of patterns in Figure 5.

Concern 4: Requests comparison against Pi0.5.

**Response:**

 We clarified compute fairness and explained why cross-entropy remains suitable for VLA learning, and we added block diffusion to the related work section. The OpenVLA-OFT results in Table 1 have been updated, and a full inference-time breakdown is now included in Table 2. We expanded the explanation of the patterns observed in Figure 5. Performance of Pi0.5 has been added to the revised paper (Table 1 and Figure 9).

---

### Meta-Review · Area_Chair_74E5 · 2025-12-23

**Summary:**

The reviewers generally recognized the paper's contribution to the field of Vision-Language-Action (VLA) modeling, particularly in addressing the efficiency-performance trade-off through a novel learnable action tokenizer (FASTerVQ) and block-wise autoregressive (BAR) decoding. Most of the concerns stem from the insufficient ablations, inference speed, and additional experiments with baselines. During the rebuttal, I believe most of the concerns have been addressed. However, it would be great to test the tokenizer on more challenging real-world scenarios rather than simple settings. I would recommend an acceptance for the paper.

**Reviewer Concerns:**

I think most of the concerns have been addressed. The remaining issue is the failure mode discussion. The authors didn't really discuss how the method fails in real-world scenarios under more complex and challenging tasks.

**Reviewer Scores:**

I think 8Nbo would have raised his score if 8Nbo was able to participate fully in discussion, because the authors' response has mostly addressed 8Nbo's concerns. Other reviews will likely keep their original score.

---

### Decision · Program_Chairs · 2026-01-26

Accept (Poster)